# Synergistic actions of v-SNARE transmembrane domains and membrane-curvature modifying lipids in neurotransmitter release

**Madhurima Dhara[1], Maria Mantero Martinez[1], Mazen Makke[1], Yvonne Schwarz[1], Ralf Mohrmann[2], Dieter Bruns[1]\***

[1]Institute for Physiology, Center of Integrative Physiology and Molecular Medicine, Saarland University, Homburg, Germany; [2]Institute for Physiology, Otto-von-Guericke University, Magdeburg, Germany

**Abstract** Vesicle fusion is mediated by assembly of SNARE proteins between opposing membranes. While previous work suggested an active role of SNARE transmembrane domains (TMDs) in promoting membrane merger (Dhara et al., 2016), the underlying mechanism remained elusive. Here, we show that naturally-occurring v-SNARE TMD variants differentially regulate fusion pore dynamics in mouse chromaffin cells, indicating TMD flexibility as a mechanistic determinant that facilitates transmitter release from differentially-sized vesicles. Membrane curvature-promoting phospholipids like lysophosphatidylcholine or oleic acid profoundly alter pore expansion and fully rescue the decelerated fusion kinetics of TMD-rigidifying VAMP2 mutants. Thus, v-SNARE TMDs and phospholipids cooperate in supporting membrane curvature at the fusion pore neck. Oppositely, slowing of pore kinetics by the SNARE-regulator complexin-2 withstands the curvature-driven speeding of fusion, indicating that pore evolution is tightly coupled to progressive SNARE complex formation. Collectively, TMD-mediated support of membrane curvature and SNARE force-generated membrane bending promote fusion pore formation and expansion.

**\*For correspondence:**
dieter.bruns@uks.eu

**Competing interests:** The authors declare that no competing interests exist.

## Introduction

Membrane fusion is the key event in many cell biological processes like exocytosis, intracellular cargo trafficking and even fertilization. The core of the neuronal exocytotic machinery is composed of soluble *N*-ethylmaleimide-sensitive-factor attachment receptor proteins (SNAREs) comprising the vesicular associated membrane protein 2 (VAMP2, also known as synaptobrevin2) and the plasma membrane anchored proteins Syntaxin and SNAP-25 (*Südhof and Rothman, 2009*; *Jahn and Fasshauer, 2012*). According to current models, SNARE-mediated fusion is initiated by the formation of an hourglass-shaped membrane stalk between the interacting membranes (fusion of the proximal leaflets) that radially widens until the distal leaflets meet and rupture, forming the nascent fusion pore (*Chernomordik and Kozlov, 2003*). In line with the view that the fusion pore and its preceding intermediates represent highly curved membrane structures, curvature-modulating lipid components have been shown to enhance fusion of protein-free lipid bilayers, possibly by lowering the energy barriers of the high-curvature intermediates (*Chernomordik and Kozlov, 2008*). Similarly, application of curvature-promoting phospholipids or their de novo generation by phospholipase D has been shown to regulate reconstituted SNARE-mediated liposome fusion (*Tong et al., 2009*; *Kreutzberger et al., 2017*) and Ca$^{2+}$-triggered exocytosis (*Amatore et al., 2006*; *Churchward et al., 2008*; *Roth, 2008*; *Zhang and Jackson, 2010*).

Fusion pore dynamics are not only crucial for regulating vesicle recycling but also modulate the kinetics and the extent of cargo release (for review see *Wu et al., 2017*). Resealing fusion pores allow for rapid recapture of intact vesicles, whereas complete membrane merger requires the de novo generation of the organelle. Since many cells co-package small and large cargo molecules in their secretory vesicles, the initial narrow pore can act as a molecular sieve causing size dependent cargo retention. In pancreatic β-cells, non-expanding fusion pores allow for release of small cargo molecules such as ATP, but hinder expulsion of insulin (*MacDonald et al., 2006*), a scenario that contributes to the development of type 2 diabetes (*Collins et al., 2016*; *Xu et al., 2017*; *Guček et al., 2019*). Adrenal chromaffin cells release small catecholamines through flickering small pores, and can additionally release larger cargo in an activity-dependent manner (*Fulop et al., 2005*). Moreover, fusion pore dynamics may even affect release of neurotransmitters and the mode of endocytosis during synaptic vesicle fusion (*Alabi and Tsien, 2013*). Yet, despite the significance of pore dynamics for efficient neurotransmitter or hormone release, mechanisms controlling fusion pore expansion are still poorly understood. Even the very nature of the nascent fusion pore, being either lipidic or proteinaceous, is under debate (*Jackson and Chapman, 2006*; *Bao et al., 2016*).

Previously, we have shown that the VAMP2 TMD plays an active role in membrane fusion, catalyzing fusion initiation and fusion pore expansion at the millisecond time scale (*Dhara et al., 2016*). Our functional analysis supported the view that overall structural flexibility of the TMD, as promoted by the number of ß-branched amino acids (like valine or isoleucine), rather than specific residues within the VAMP2 TMD, determines the exocytotic response (*Dhara et al., 2016*; *Han et al., 2016*). Furthermore, molecular dynamics simulations of v-SNARE TMDs embedded in an asymmetric membrane (to mimic the physiological lipid composition of synaptic vesicles) revealed that mutant variants with a poly-leucine or with a poly-valine TMD decrease or increase the root mean square fluctuation (RMSF) of the backbone atoms for the peptide, respectively (*Dhara et al., 2016*). In the same line, sequence-specific back bone dynamics of isolated TMD model helices (probed by hydrogen/deuterium exchange) enhanced the fusogenicity of liposomes in in vitro assays (*Stelzer et al., 2008*; *Quint et al., 2010*), pointing to an active role of the v-SNARE TMD in the fusion mechanism. Overall, these results showed that the function of v-SNARE TMDs clearly goes beyond simple membrane anchoring, but left the mechanisms of the underlying protein-lipid interplay unclear.

Here, we set out to elucidate key aspects of the effect of v-SNARE transmembrane domains on $Ca^{2+}$-triggered exocytosis. By using a combination of membrane capacitance measurements and photolytic $Ca^{2+}$-uncaging as well as carbon fiber amperometry, we followed vesicle exocytosis in chromaffin cells from pre- to postfusional stages. Our results show that naturally occurring v-SNARE TMD variants with higher or lower content of ß-branched amino acids than in the VAMP2 protein differentially affect fusion pore dynamics, suggesting a general physiological significance of TMD flexibility in membrane fusion and transmitter discharge. Membrane-incorporated lipids like lysophosphatidylcholine (LPC) or oleic acid (OA) affected fusion induction and subsequent pore expansion in a membrane leaflet-specific fashion, correlating with their intrinsic curvature preference of the cytoplasmic and extracellular leaflet in the context of highly bent fusion intermediates. These results indicate that membrane mechanics represent a rate-limiting energy barrier for $Ca^{2+}$-triggered fusion of chromaffin granules, which proceeds via the formation of a membrane stalk intermediate into a lipidic fusion pore. Importantly, slowing of fusion pore expansion by a v-SNARE variant with a rigid TMD was fully rescued by either intracellular OA or extracellular LPC, indicating that v-SNARE transmembrane anchors and phospholipids cooperate in membrane remodeling by supporting membrane curvature at the fusion pore neck.

## Results

### Naturally occurring v-SNARE TMD variants regulate fusion pore dynamics without affecting overall secretion

Starting point of our study was the observation that v-SNARE isoforms driving fusion of differentially-sized secretory vesicles contain highly variable amounts of ß-branched amino acids within the N-terminal half of their TMDs (*Dhara et al., 2016*).

Isoform usage in different tissues indicates that VAMP variants with a low number of ß-branched amino acids in this TMD region (e.g. VAMP1, 22% ß-branched amino acids) are preferentially involved in fusion of small synaptic vesicles, whereas isoforms with high content of ß-branched amino acid (e.g. VAMP8, 77% ß-branched amino acids) are responsible for fusing larger sized zymogen granules or mast cell vesicles (*Figure 1A*). As the area of highly curved membrane within the fusion pore increases with vesicle size and membrane bending is thought to oppose fusion pore expansion (*Zhang and Jackson, 2010*; *Kawamoto et al., 2015*), it stands to reason that v-SNARE variants with a higher number of ß-branched amino acids within their TMDs represent a necessary functional adaptation to ensure rapid fusion pore expansion and *bona fide* cargo release. To test this hypothesis we generated chimeric mutants of VAMP2 by exchanging its TMD with that of either VAMP8 or with VAMP1 (denoted VAMP2-VAMP8TMD and VAMP2-VAMP1TMD, *Figure 1A*). Secretion was determined with simultaneous membrane capacitance (CM) measurements and carbon fiber amperometry in response to continuous intracellular perfusion with solution containing 19 µM free calcium. VAMP2 and its mutant variants were comparatively analyzed in a gain-of-function approach by viral expression on the genetic null background of *VAMP2/VAMP3* double knock-out (dko) chromaffin cells, which are devoid of any exocytosis (*Borisovska et al., 2005*). The expression of VAMP2-VAMP8TMD rescued total secretion like the wild type (wt) protein, as indicated by a comparable capacitance increase and similar amperometric event frequency (*Figure 1B*). Yet, analysis of amperometric spike wave forms demonstrated that the mutant protein had a profound impact on the kinetics of transmitter discharge. This gain-of-function phenotype is characterized by significantly higher amplitudes and faster kinetics of the main amperometric spike when compared with VAMP2-mediated fusion events (*Figure 1C,D*). Moreover, the VAMP2-VAMP8TMD mutant also significantly shortened the prespike signal and increased its current fluctuations (*Figure 1E*), which report transient changes in neurotransmitter flux through the early fusion pore (*Kesavan et al., 2007*). Analysis of the root-mean-square (rms) noise of the prespike's current derivative, serving as a threshold-independent parameter of fusion pore fluctuations, corroborated that VAMP2-VAMP8TMD expression significantly enhances the fusion pore jitter (*Figure 1E*). In contrast, expression of the VAMP2-VAMP1TMD chimera with only 22% ß-branched amino acid content in the N-terminal half of the TMD significantly slowed down catecholamine release from chromaffin granules (indicated by lower spike amplitudes, increased rise-times and half-width values, *Figure 1C–E*) without changing the overall rate of fusion events (*Figure 1B*). Immunofluorescence analyses revealed nearly similar expression levels of VAMP2 and its mutant proteins in dko cells (*Figure 1—figure supplement 1A, B*). Using high resolution structured illumination microscopy we found that the VAMP2 TMD mutants are sorted to large dense core vesicles with similar efficiency like the wt protein (*Figure 1—figure supplement 1C–E*), attributing the observed fusion deficits to changes in TMD-mediated function.

Moreover, photolytic uncaging of intracellular [Ca]i evoked similar synchronous secretion of dko cells expressing VAMP2 or its chimeric mutants (*Figure 2A,C*). Both release components of the exocytotic burst, the rapidly releasable pool (RRP) and the slowly releasable pool (SRP), as well as the sustained rate of secretion were unaltered (*Figure 2B,D*, left panels). Furthermore, we found no indication for changes in the stimulus-secretion coupling (*Figure 2B,D*, right panels), rendering the possibility unlikely that TMD-mutants destabilizes the membrane-proximal part of the SNARE complex which should affect synaptotagmin binding (*Dai et al., 2007*) and, thereby, exocytosis timing.

Taken together, these results extend our original finding that the content of ß-branched amino acids crucially determines conformational properties of the VAMP2 TMD governing the fusion process from opening of the nascent fusion pore to its final expansion (*Dhara et al., 2016*). They show that not only artificial TMD mutants (carrying either helix-stabilizing leucines or flexibility–promoting ß-branched isoleucine/valine residues) but also naturally–occurring variants of v-SNARE TMDs specifically alter fusion pore dynamics without changing overall secretion or stimulus-secretion coupling. Importantly, substitution of the VAMP2 TMD with either the VAMP1 or the VAMP8 TMD caused correlated changes in spike waveform, even producing a gain-of-function phenotype in pore expansion kinetics for the VAMP8 TMD enriched in ß-branched residues. The latter phenotype of a faster catecholamine secretion for VAMP2 with the VAMP8 TMD provides strong evidence that adapting the structural flexibility of v-SNARE TMDs to the vesicle size facilitates the expansion of the fusion pores and, thus, serves the release of bulky cargo molecules from large-sized zymogen granules.

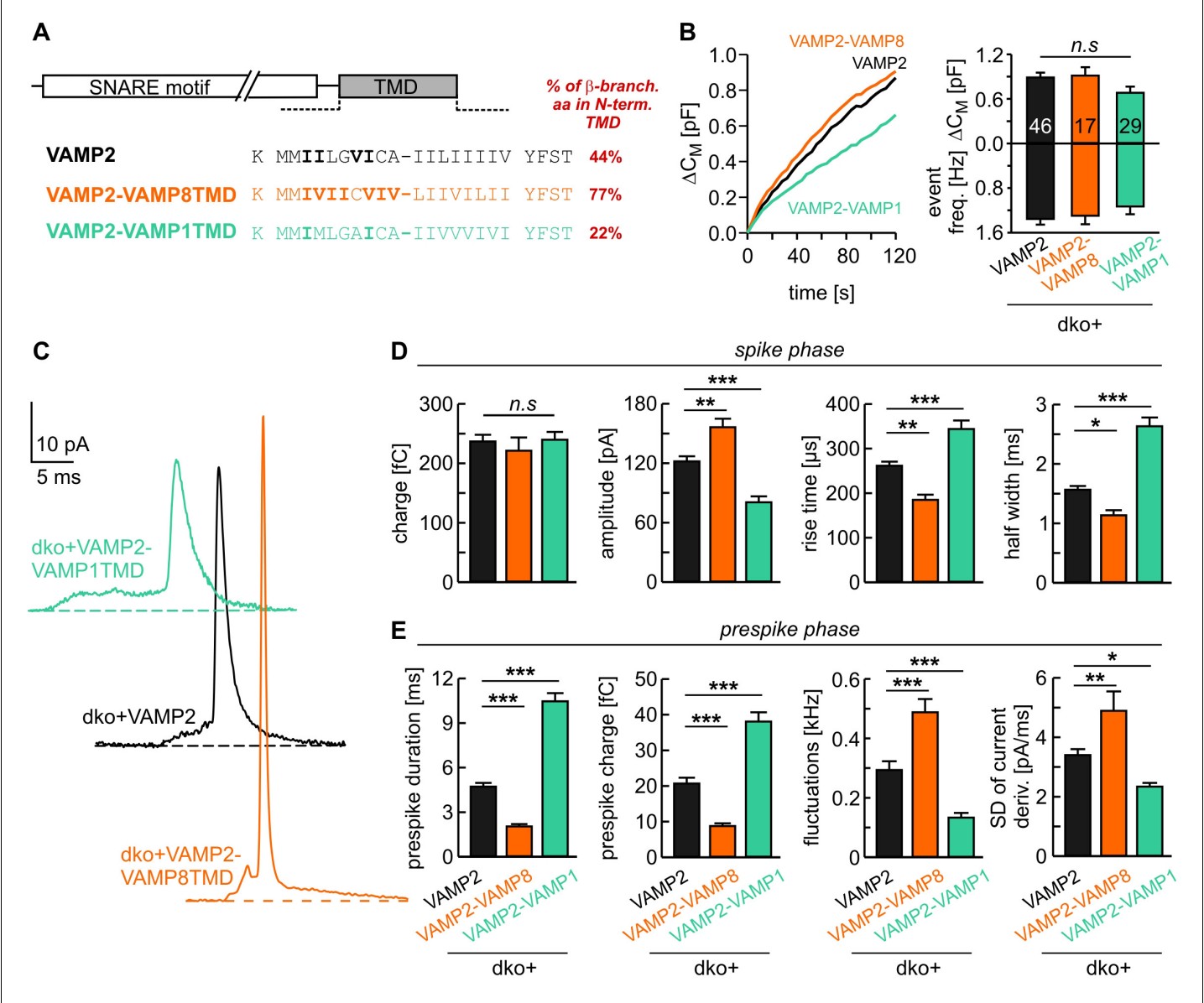

**Figure 1.** v-SNARE TMD variants differentially control fusion pore kinetics. (A) Primary sequence of VAMP2 TMD and its chimeras with a VAMP8 or VAMP1 TMD. ß-branched residues of the N-terminal TMD region are highlighted in bold. (B) Mean capacitance changes in response to intracellular perfusion with 19 µM free $Ca^{2+}$ in the indicated groups (left panel). Total ΔCM (top) and amperometric event frequency (bottom) measured over 120 s (right panel) show that both v-SNARE chimeric variants support normal exocytosis. $T_0$ is the first time point of CM measurement, 2–3 s after starting the $Ca^{2+}$-infusion via the patch pipette. (C) Exemplary amperometric events with similar charge but altered release profiles for the indicated VAMP2 variants. (D, E) VAMP2-VAMP8TMD mutant shortens the prespike duration and accelerates the spike waveform (increased amplitude, reduced 50–90% rise time, and half width). Conversely, the VAMP2-VAMP1TMD prolongs the prespike phase, slows down the kinetics of the spike and reduces its amplitude. Values are given as mean of median determined from the indicated parameter's frequency distribution for each cell. Data were collected from cells/events measured for VAMP2 (46/3588), VAMP2-VAMP8TMD (17/1232), VAMP2-VAMP1TMD (29/1757). Only cells with >20 events were considered. Data are presented as mean ± SEM. *p<0.05, **p<0.01, ***p<0.001, one-way analysis of variance followed by Tukey-Kramer post test. The online version of this article includes the following figure supplement(s) for figure 1:

**Figure supplement 1.** VAMP2-TMD mutants exhibit a similar expression and sorting pattern to granules as the wt protein.

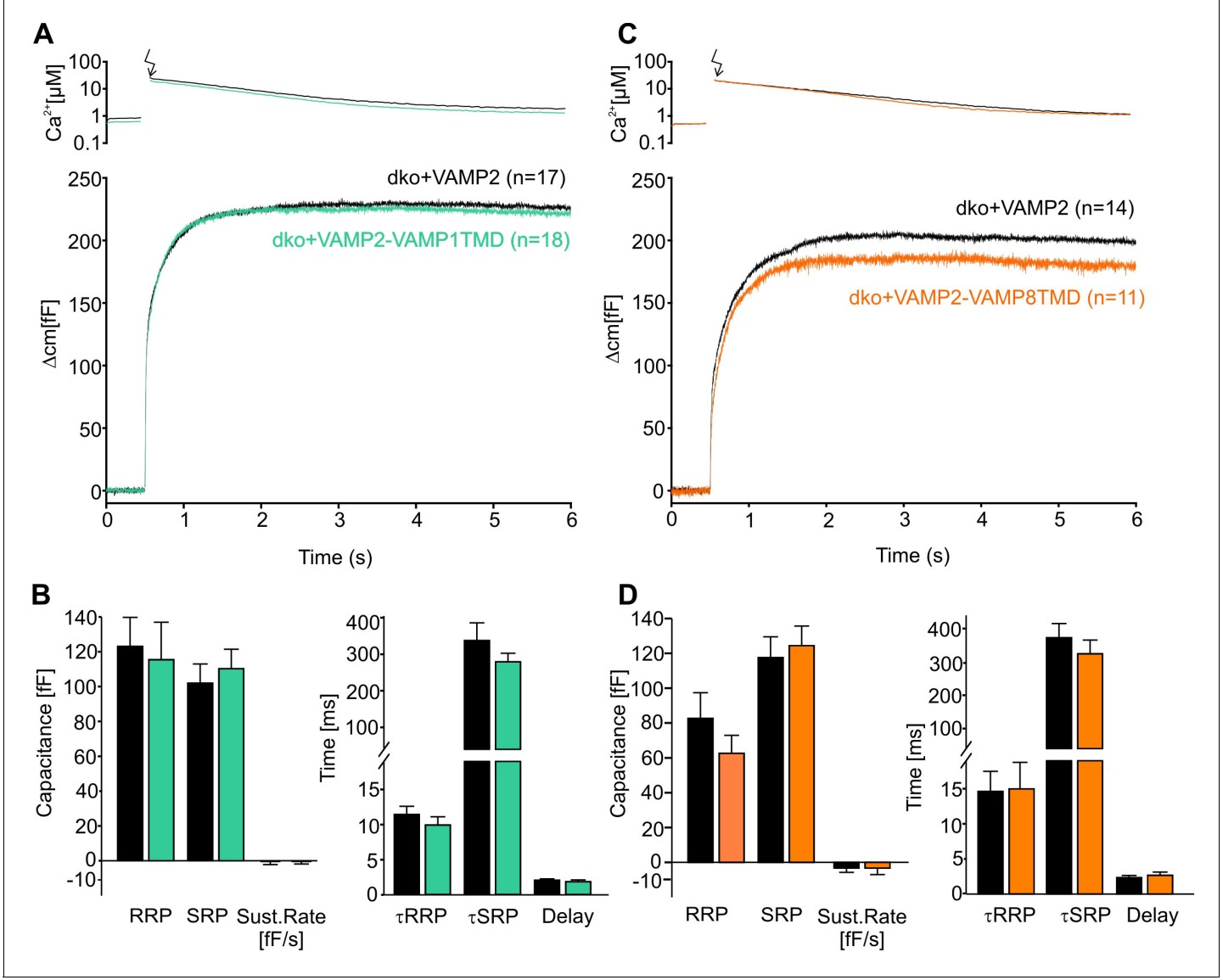

**Figure 2.** The naturally occurring v-SNARE TMD variants do not affect synchronous secretion. (**A**) Mean flash-induced $[Ca^{2+}]i$ levels (top panel) and corresponding CM responses (lower panels) of dko cells expressing either VAMP2 or the VAMP2-VAMP1TMD mutant. (**B**) The VAMP2-VAMP1TMD mutant fully restored the flash evoked response with an unchanged RRP, SRP and sustained rate of release. Neither the kinetics of release τRRP, τSRP nor the secretory delay was altered. (**C**) Mean flash-induced $[Ca^{2+}]i$ levels (top panel) and corresponding CM responses (lower panels) of dko cells expressing either VAMP2 or the VAMP2-VAMP8TMD. (**D**) Neither the extent (left panel) nor the kinetics of secretion (right panel) were altered with expression of the VAMP2-VAMP8TMD. Arrow indicates UV-flash. Data are represented as mean ± SEM and numbers of cells are indicated within brackets. Mann-Whitney Rank Sum test.

## Curvature-modifying phospholipids affect fusion in a leaflet-specific fashion

We next asked how v-SNARE TMDs may control the kinetic behavior of the expanding fusion pore. Recent molecular dynamics simulations showed that isolated TMDs of VAMP2 or syntaxin drastically lower the free energy of the metastable membrane stalk, an observation that was related to the local thinning of the membrane (negative hydrophobic mismatch) imposed by the TMDs (*Smirnova et al., 2019*).The observed mutual attraction between the stalk and TMDs accumulates the TMDs at the stalk base (*Smirnova et al., 2019*), a scenario that may generate local (negative) curvature which eases membrane bending and pore growth. To explore this idea, we extracellularly applied lipid molecules that confer either positive (lysophosphatidyl choline, LPC) or negative (oleic acid, OA)

membrane curvature (*Figure 3A*). We found that neither LPC nor OA (2 µM) altered the total secretion compared to controls in experiments testing tonic secretion (*Figure 3B*) or synchronized exocytosis (photolytic uncaging of intracellular $Ca^{2+}$, *Figure 3—figure supplement 1*). However, extracellular application of LPC strongly accelerated the kinetics of cargo release from individual vesicles, as indicated by higher amplitudes and faster kinetics of the amperometric spikes (*Figure 3C–E*). OA, instead, slowed down the overall amperometric current indicating decelerated catecholamine release and slower fusion pore expansion. Similar differences between LPC and OA were also seen for the current fluctuations of the prespike signal (fluctuation frequency in kHz (mean ± sem, one-way analysis of variance vs. wt+DMSO): wt+DMSO 0.31 ± 0.02, wt+LPC 0.56 ± 0.027 (p<0.001), wt+OA 0.13 ± 0.019 (p<0.001); SD noise (pA/ms): wt+DMSO 3.73 ± 0.18, wt+LPC 6.05 ± 0.38 (p<0.001), wt+OA 2.5 ± 0.14 (p<0.005)). These differential effects of cone and inverted-cone shaped lipids are consistent with a net positive curvature of the extracellular leaflet within the open fusion pore (*Chernomordik and Kozlov, 2008*). Taken together, LPC and OA oppositely affect both, the prespike and the spike phase of transmitter discharge, implying that membrane bending imposes a rate-limiting energy barrier on fusion pore expansion.

Given that the cytoplasmic leaflet of potential fusion intermediates like the hemifusion stalk or the fusion pore are characterized by a net negative curvature (*Chernomordik and Kozlov, 2008*), we next asked how lipid molecules (5 µM) infused into cells via the patch pipette may alter secretion. Intriguingly, cone-shaped OA molecules clearly enhanced tonic secretion, whereas positive curvature inducing LPC suppressed release (*Figure 4A,B*). In the same line, OA enhanced the synchronized, flash-evoked response upon photolytic $Ca^{2+}$-uncaging and LPC lowered it (*Figure 4—figure supplement 1*). The stimulus-secretion coupling was unchanged, indicating that curvature-modifying lipids do not interfere with the molecular steps underlying fusion triggering, but alter the number of fusion-competent vesicles. Moreover, intracellular OA accelerated the kinetics of catecholamine release from single vesicles during prespike and spike phase, while intracellular LPC severely slowed down the cargo discharge (*Figure 4C–E*). These results are in excellent agreement with previous findings studying protein-free liposome or viral membrane fusion (*Chernomordik and Kozlov, 2003*; *Melia et al., 2006*), indicating that $Ca^{2+}$-triggered exocytosis transits through similar highly-curved membrane intermediates (i.e. hemifusion stalk) *en route* to complete membrane merger (*Zhao et al., 2016*). Yet, the observed dependency of fusion rates on intracellular LPC and OA is difficult to reconcile with concepts that exocytosis begins with a proteinaceous fusion pore (*Zhang and Jackson, 2010*). They rather suggest that curvature-accommodating phospholipids facilitate stalk formation, catalyze the transition to pore opening and promote pore enlargement, results which are clearly in accord with the continuum 'stalk-pore' model of membrane fusion (*Chizmadzhev et al., 1995*; *Chizmadzhev et al., 2000*; *Chernomordik and Kozlov, 2003*). It is important to note that, lipids with a favorable curvature do not simply stabilize the exocytotic fusion pore, but even more strongly lower the energy barrier for further pore growth (*Chizmadzhev et al., 2000*).

## Curvature-inducing phospholipids rescue fusion deficits of the v-SNARE TMD mutant

Application of curvature-generating phospholipids like extracellular LPC (or intracellular OA) and expression of the VAMP2-VAMP8-TMD caused remarkably congruent alterations in the kinetics of single fusion events (compare *Figures 1* and *3*), pointing to the possibility that TMD flexibility and membrane curvature manipulation target the very same mechanism of pore evolution. To further address this point, we comparatively analyzed the effects of extracellular LPC application on secretion of dko cells expressing either VAMP2 or the VAMP2polyL mutant protein, which contains helix-stabilizing leucine residues within the TMD region (*Dhara et al., 2016*). Consistent with our previous work, the helix-rigidifying TMD mutation strongly impaired exocytosis (*Figure 5A,B*) and decelerated fusion pore dilation (*Figure 5C,D*). While extracellular LPC application failed to alter the overall rate of tonic secretion of dko cells expressing VAMP2polyL (*Figure 5A,B*), the same treatment indeed fully rescued the kinetic deficits of transmitter release seen in the amperometric spike waveform. Both phases in the release time course form single vesicles, the prespike and the spike phase, were similarly affected (*Figure 5C,D*). Thus, LPC speeds up cargo release and levels out any differences in fusion pore expansion between wild type and mutant protein suggesting common mechanisms of action for v-SNARE TMDs and phospholipids.

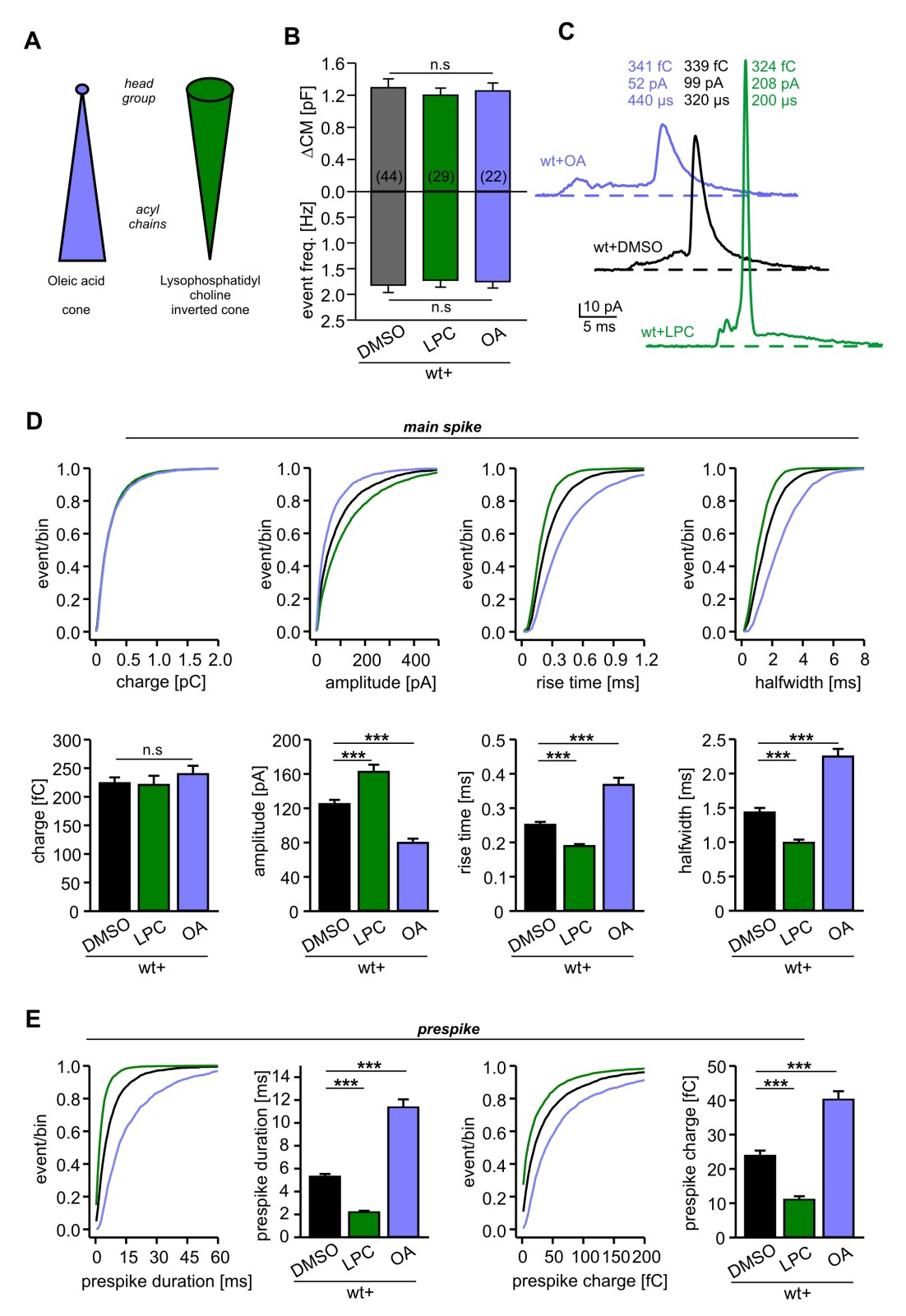

**Figure 3.** Extracellular application of cone and inverted cone shaped lipid molecules differentially controls fusion pore kinetics. (**A**) Schematic representation of oleic acid (OA, cone) and lysophosphatidyl choline (LPC, inverted cone) inducing negative and positive curvature of the membrane, respectively. (**B**) Extracellular application of LPC or OA does not affect the average ΔCM (top) or the amperometric event frequency (bottom). Data are averaged from the indicated number of cells. (**C**) Exemplary amperometric events with similar charge but altered kinetic release profile for the indicated

*Figure 3 continued on next page*

*Figure 3 continued*

conditions. (D) Cumulative frequency distributions (upper panels) and cell-weighted averages (lower panels) for the indicated parameter of the main spike. Note that LPC accelerates the spike waveform, whereas OA slows down the current transient. (E) Extracellular LPC and OA oppositely affect the prespike phase. LPC shortens the prespike duration and reduces the prespike charge, while OA prolongs the prespike duration and increases its charge. Values are given as mean of median determined from the indicated parameter's frequency distribution for each cell. Data were collected from cells/events measured for wt+DMSO (44/4322), wt+LPC (29/3554), wt+OA (22/2616). Only cells with >20 events were considered. Data are represented as mean ± SEM. ***p<0.001, one-way analysis of variance followed by Tukey-Kramer post test.

The online version of this article includes the following figure supplement(s) for figure 3:

**Figure supplement 1.** Extracellular application of LPC or OA does not affect overall secretion.

In contrast to extracellular LPC application, infusion of OA restored - at least in part - the impaired fusion rate seen with VAMP2polyL (*Figure 6A–C*). The relative increase in total secretion upon OA infusion was significantly higher in VAMP2polyL expressing cells than those rescued with wt VAMP2 (*Figure 6C*). This indicates that the fusion facilitating action of OA can partly compensate for the impaired VAMP2 mutant function, suggesting that negative curvature is specifically required for the formation of a stalk intermediate between the contacting proximal monolayers of the vesicle and the plasma membrane. Moreover, OA fully rescued the slowed discharge of catecholamine from single vesicles observed with the VAMP2polyL mutant (*Figure 6D,E*), similar to the action of extracellular LPC (*Figure 5C,D*). OA also restored the reduced current fluctuations of the prespike signal with VAMP2polyL (fluctuation frequency in kHz (mean ± sem, one-way analysis of variance vs. VAMP2+DMSO): VAMP2+DMSO 0.28 ± 0.02, polyL+DMSO 0.12 ± 0.019 (p=0.03), VAMP2+OA 0.56 ± 0.05 (p<0.001), polyL+OA 0.61 ± 0.04 (p<0.001); SD noise (pA/ms): VAMP2+DMSO 3.45 ± 0.22, polyL+DMSO 2.21 ± 0.14 (p=0.027), VAMP2+OA 5.47 ± 0.4 (p<0.001), polyL+OA 5.39 ± 0.35 (p<0.001)). Overall, these results show that lipid molecules with a favorable shape for either leaflet fully compensate for the functional deficits of the rigid TMD in VAMP2polyL during fusion pore expansion, indicating that conformational flexibility of the v-SNARE TMD primarily regulates fusion pore growth by affecting membrane curvature.

## Extracellular LPC fails to rescue fusion deficits caused by impaired SNARE force

To evaluate the specificity of curvature-generating lipids for rescuing an altered TMD-membrane interplay in fusion pore expansion, we also investigated the ability of extracellular LPC to compensate for other manipulations known to affect pore growth. Previously, we have shown that expression of complexin II (CpxII) not only hinders tonic exocytosis of chromaffin granules, but also slows down the expansion of the nascent fusion pore, a phenotype which could be related to interference with final SNARE assembly (*Dhara et al., 2014*; *Makke et al., 2018*). Confirming our original observations, CpxII expression significantly prolonged the duration of the prespike without changing the properties of the main spike (*Figure 7C,D*). CpxII also reduced the frequency of the prespike's current fluctuations, which likely reflect attempts of the SNARE machinery to widen the initial fusion pore (*Kesavan et al., 2007*). Intriguingly, application of extracellular LPC failed to rescue the strong prespike phenotype of CpxII overexpression, although a small decrease in the duration of the prespike signal was still observed (*Figure 7D*). Furthermore, LPC treatment did not change the overall rate of secretion (*Figure 7A,B*) like in wt cells (*Figure 3B*). Yet, LPC clearly replicated other effects by increasing the amplitude of the main spike and accelerating its kinetics (*Figure 7C*), which proved similar treatment like in our earlier experiments (compare with *Figure 3D*).

The combined set of data shows that curvature-mediating phospholipids do not generally accelerate $Ca^{2+}$-triggered exocytosis, but specifically compensate for the functional deficits of rigid v-SNARE TMDs. Only interventions that actually target the same step or fusion intermediate are expected to lift the inhibition and produce a substantial compensation. CpxII expression, on the other hand, makes the final association of SNARE proteins a rate-limiting step that hinders the curvature-driven acceleration of nascent fusion pore expansion. Thus, fusion pore expansion (and membrane bending) can only proceed in concert with progressive zipping of the membrane-anchored SNARE pins. In summary, our results show that the TMDs of v-SNARE proteins control key properties

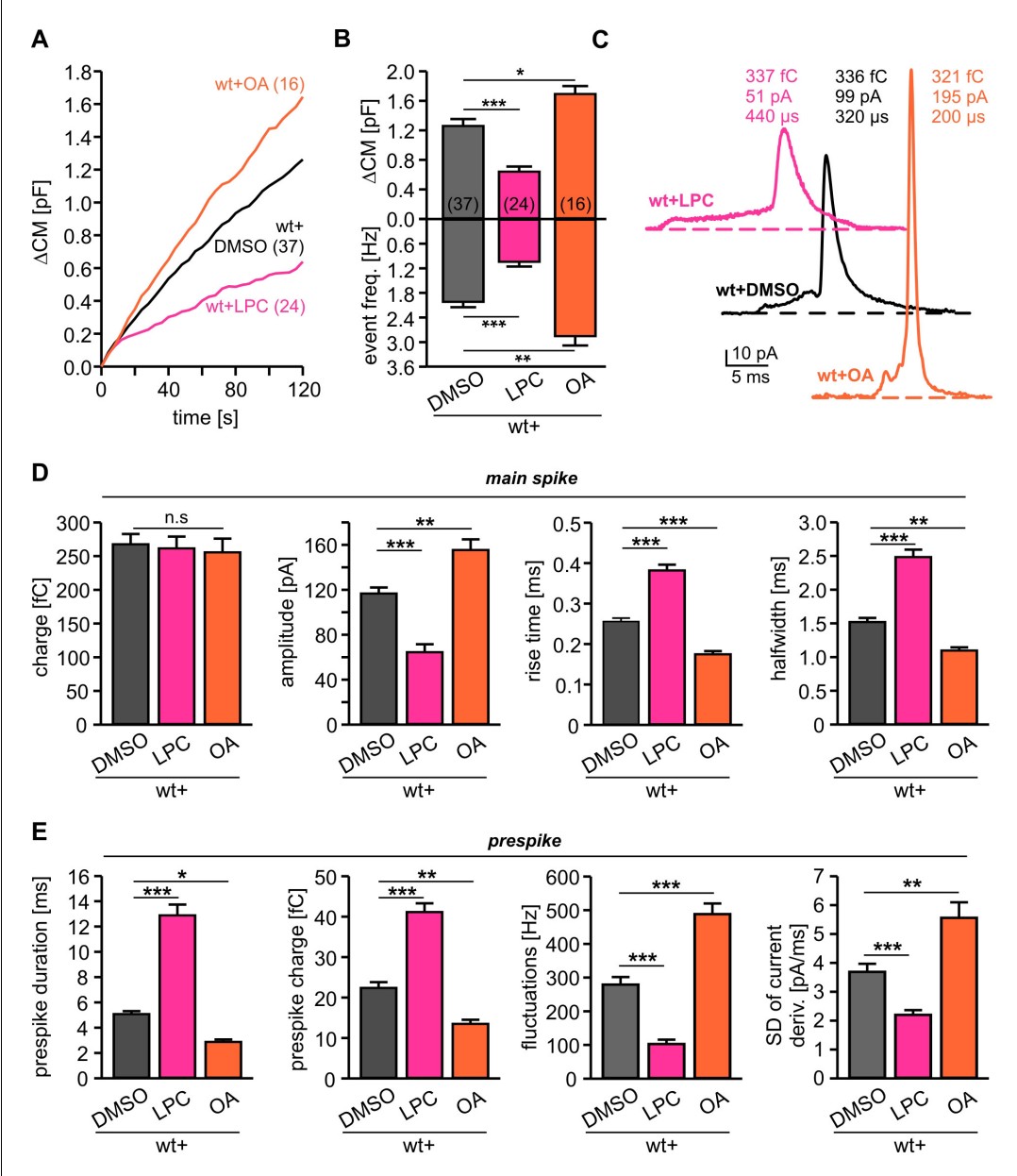

**Figure 4.** Intracellular application of LPC and OA oppositely affects total membrane fusion and fusion pore kinetics. (**A**) Mean capacitance changes in response to stimulus for the indicated groups. (**B**) Intracellular LPC reduces and intracellular OA enhances total ΔCM response (top) and the corresponding amperometric event frequency (bottom). Data are averaged from the indicated number of cells. (**C**) Exemplary amperometric events with similar charge but altered release kinetic profile for wt+DMSO (control), wt+LPC and wt+OA. (**D**) Intracellular LPC decelerates the spike waveform by lowering the amplitude, prolonging the rise time and half width (without affecting the spike charge), whereas OA accelerates the current transient. (**E**) Intracellular LPC and OA oppositely affect the indicated parameters of the prespike phase. Values are given as mean of median determined from the indicated parameter's frequency distribution for each cell. Data were collected from cells/events measured for wt (37/4023), wt+LPC (24/1674), wt+OA (16/2104). Only cells with >20 events were considered. Data are represented as mean ± SEM. *p<0.05, **p<0.01, ***p<0.001, one-way analysis of variance followed by Tukey-Kramer post test.

The online version of this article includes the following figure supplement(s) for figure 4:

**Figure supplement 1.** Intracellular application of LPC or OA affects synchronous secretion.

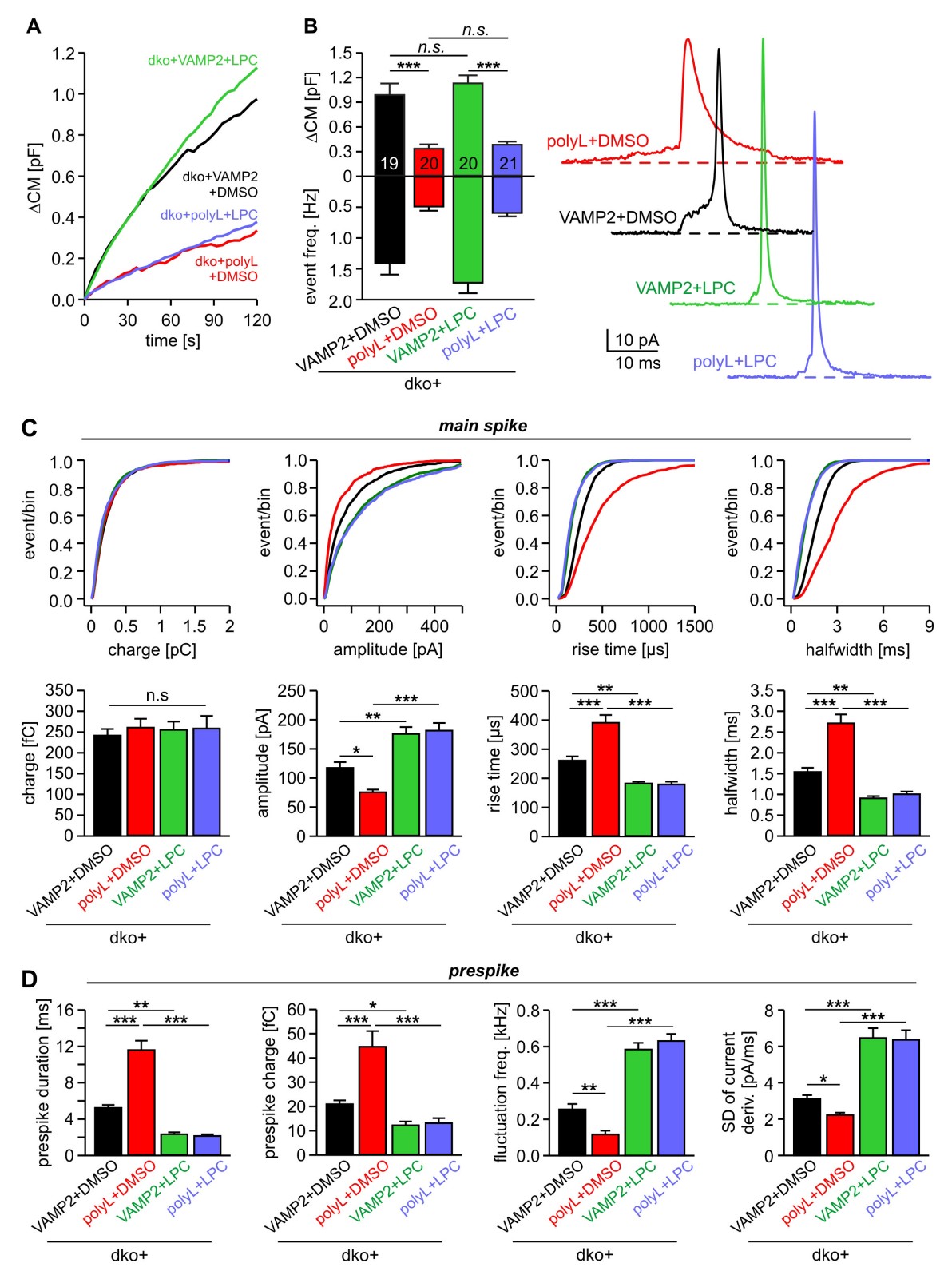

**Figure 5.** Extracellular LPC overrides the different fusion pore kinetics of VAMP2 and its TMD mutant. (**A**) Mean capacitance changes in response to stimulus for the indicated groups. (**B**) Total ΔCM (top) and amperometric event frequency (bottom) show that VAMP2polyL reduces exocytosis, which is unaffected by extracellular LPC application. Data are averaged from the indicated number of cells. Exemplary amperometric events with similar charge but altered release kinetic profile for the indicated group (right panel). (**C, D**) Cumulative frequency distributions and cell-weighted averages for the

*Figure 5 continued on next page*

*Figure 5 continued*

indicated parameters of the spike (C) and prespike phase (D) of single fusion events. The VAMP2polyL mutant slows the overall spike and prespike kinetics. Extracellular LPC strongly accelerates the cargo discharge kinetics in VAMP2 and VAMP2polyL expressing cells and overrides the differences in fusion pore expansion between wt and mutant protein. Note that the similar 50–90% spike rise time of wt and VAMP2polyL is not due to signal filtering (2 kHz), which allows for minimum rise times of about 83 µs. Values are given as mean of median determined from the indicated parameter's frequency distribution for each cell. Data were collected from cells/events measured for dko+VAMP2 (19/1315), dko+VAMP2polyL (20/705), dko+VAMP2+LPC (20/1698) and dko+VAMP2polyL+LPC (21/882). Only cells with >20 events were considered. Data are represented as mean ± SEM. *$p<0.05$, **$p<0.01$, ***$p<0.001$, one-way analysis of variance followed by Tukey-Kramer post test.

of the actual membrane fusion step by significantly reducing the energy needed to bend the adjacent membrane within the highly curved fusion pore neck.

## Discussion

The most probable pathway to membrane fusion does not only involve specific protein-protein interactions but also requires optimized protein-lipid interactions. In the present work, we show for the first time that naturally occurring TMD variants of different v-SNARE isoforms have a decisive influence on the rate of fusion pore expansion. Given the increasing resistance to fusion pore expansion from small to large vesicles, adaptation of TMD variants with distinct conformational flexibility likely ensures that *bona fide* transmitter release can take place with similar efficiency from differentially sized vesicles in various secretory systems. Using curvature-inducing agents together with the expression of defined VAMP2 mutants, we show that favorable curvature generation in either leaflet fully restores the functional deficits of rigidifying the v-SNARE TMD helix. These results provide strong evidence that structurally flexible v-SNARE TMDs crucially impact on exocytotic pore dynamics by promoting favorable curvature to the fusing membranes (*Figure 8*).

### v-SNARE TMD variants differentially regulate membrane fusion

The transmembrane domains of SNARE proteins have long been regarded as passive elements in the membrane fusion process, serving only as anchors for the force transduction of SNARE complex formation on the membranes to be fused. Recently, we and others have shown that v-SNARE TMDs also play an autonomous, active role in exocytotic membrane fusion (*Chang et al., 2016*; *Dhara et al., 2016*; *Hastoy et al., 2017*). Specifically, our data suggested that the presence of ß-branched amino acids and the resulting structural flexibility of the VAMP2 TMD are crucial for promoting fusion pore expansion (*Dhara et al., 2016*). The different content of ß-branched amino acids in the naturally occurring TMD variants of v-SNARE proteins raised the question of whether those structural properties are exploited in facilitating the release of cargo molecules from different types of secretory vesicles. To investigate this question, we generated chimeric proteins consisting of VAMP2's cytoplasmic domain and the TMD of VAMP8 or VAMP1 and comparatively analyzed their effects on catecholamine release. Neither the decrease nor the increase of ß-branched amino acids within the N-terminal half of the TMDs of VAMP1 and VAMP8 led to proportional changes in the secretion response when compared with VAMP2. For the VAMP2-VAMP1 TMD, it only slightly reduced the rate of tonic secretion in response to $Ca^{2+}$-infusion (*Figure 1A,B*) and did not affect the synchronized, flash-evoked response (*Figure 2*). Thus, only large and systematic changes in the number of helix-rigidifying amino acids, like in the VAMP2polyL mutant, are able to interfere with exocytosis initiation. In contrast, VAMP2-VAMP8 TMD mutant allowed for much faster release than the other chimera carrying only a few ß-branched amino acids in the N-terminal half of its TMD (VAMP2-VAMP1 TMD, *Figure 1*). Thus, naturally occurring TMD variants of v-SNARE proteins do differentially affect the kinetics of transmitter release confirming an active role the TMD in the expansion of exocytotic fusion pores. In the physiological context, such a mechanism may represent an important functional adaption for tipping the balance between an expanding or non-expanding fusion pore. For VAMP8-mediated exocytosis, it ensures efficient discharge of bulky cargo molecules (e.g. hexosaminidase) from large-sized mast cell vesicles (*Lippert et al., 2007*). For VAMP-1 mediated SSV exocytosis at the NMJ (*Liu et al., 2011*), it allows release of small acetylcholine molecules as well as rapid recycling of SSVs by reducing the likelihood of complete merger with the plasma membrane. In comparison to the VAMP2polyL mutant, naturally occurring TMD variants change exocytosis

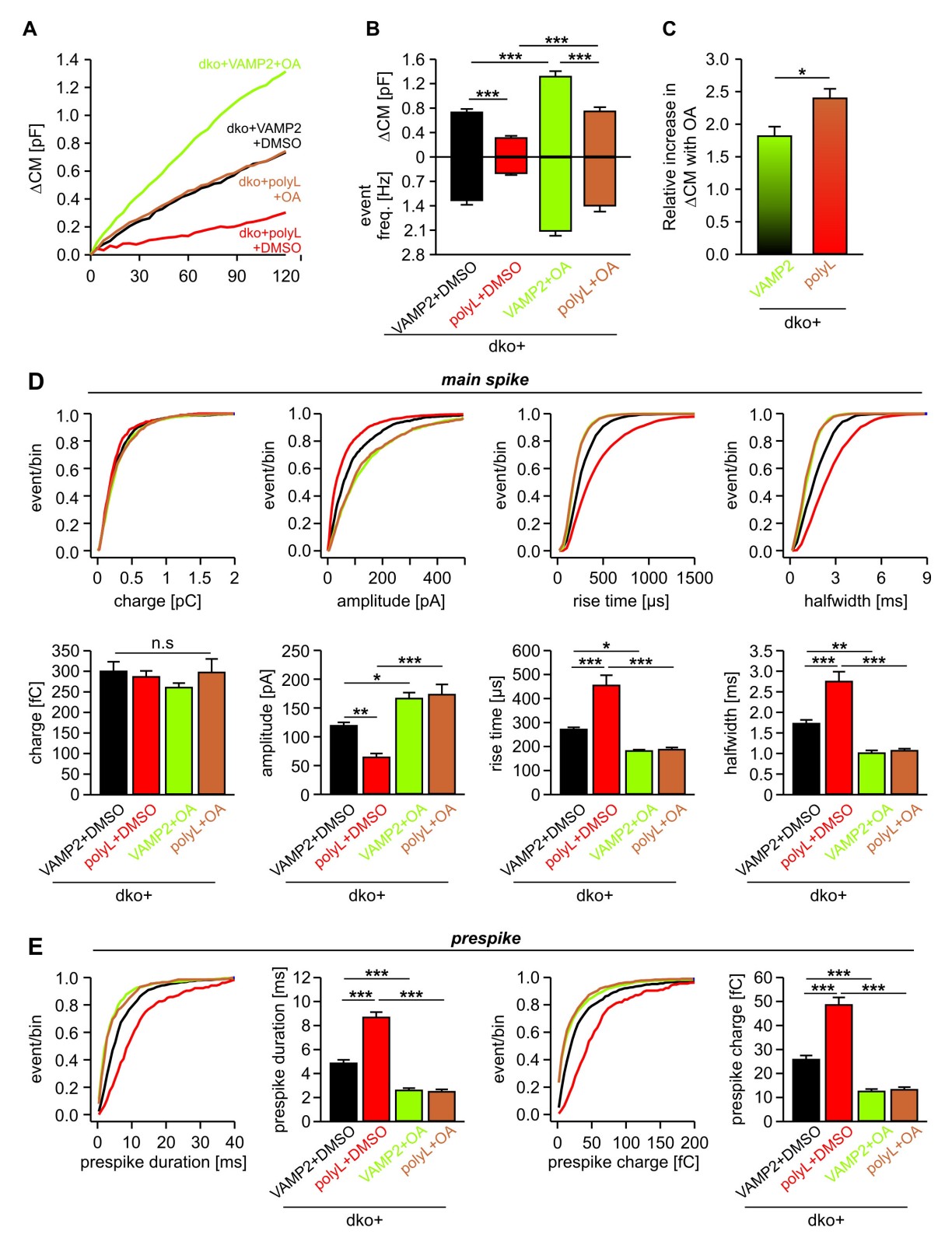

**Figure 6.** Intracellular OA enhances total exocytosis and reverses the slower release kinetics seen with polyL-TMD mutant. (**A**) Mean capacitance changes in response to stimulus for the indicated groups. (**B**) Total ΔCM (top) and amperometric event frequency (bottom) reconfirm that polyL mutant reduces exocytosis. Intracellular application of OA increases secretion both in VAMP2 and polyL expressing cells. (**C**) Increase in total secretion with intracellular OA relative to DMSO is significantly higher in polyL expressing cell compared to VAMP2. (**D, E**) Cumulative frequency distributions and

*Figure 6 continued on next page*

*Figure 6 continued*
cell-weighted averages for the indicated parameters of the spike (D) and the prespike phase (E). Note that intracellular application of OA accelerates the cargo discharge kinetics in VAMP2 expressing cells and overrides the kinetic differences in transmitter release between VAMP2 and the polyL mutant. Values are given as mean of median determined from the indicated parameter's frequency distribution for each cell. Data were collected from cells/events measured for dko+VAMP2 (18/1071), dko+VAMP2polyL (17/595), dko+VAMP2+LPC (18/2080) and dko+VAMP2polyL+LPC (18/1661). Only cells with >20 events were considered. Data are represented as mean ± SEM. *p<0.05, **p<0.01, ***p<0.001, one-way analysis of variance followed by Tukey-Kramer post test.

competence (i.e. buildup of RRP/SRP) less effectively than they alter fusion pore dynamics. This behavior ensures *bona fide* transmitter release from differentially-sized vesicles without compromising their exocytosis competence. Notably, the VAMP2-VAMP8-TMD mutant does not simply restore the kinetics of catecholamine release, but produced a gain-of-function phenotype, wherein fusion pore dilation is even accelerated beyond the rate found for wt VAMP2. Thus, it is tempting to speculate that the structural properties of the different v-SNARE TMD isoforms are optimized to meet the functional needs of the physiological release process at hand.

## The role of v-SNARE TMDs in membrane fusion

The observations described above raise the important question how TMDs actually ease the induction of fusion. Exocytosis competence of secretory vesicles is likely to be accompanied by a direct approach between the membranes to be fused (*Figure 8*). Previous molecular dynamics simulations have identified the first hydrophobic encounter, in which lipid tails splay and bridge to the adjacent leaflet, as a highly energy-demanding step *en route* to fusion (*Kasson et al., 2010*; *Smirnova et al., 2010*; *Risselada et al., 2011*). Our simultaneous membrane capacitance measurements (CM) and carbon fiber amperometry showed that expression of the helix-rigidifying VAMP2polyL mutant strongly diminishes exocytosis (*Figures 5* and *6*, see also *Dhara et al., 2016*). An attractive explanation for this phenotype would be that loss of conformational flexibility within the TMD lowers the probability of lipid splay and thereby impairs fusion initiation. In fact, our functional results are in excellent agreement with recent biochemical analyses, showing that conformationally rigid TMDs promote less proximal leaflet mixing and lipid splay than flexible TMDs (*Scheidt et al., 2018*). Furthermore, molecular dynamics simulations using an unbiased 'string method', to determine the minimum free energy path of fusion intermediates, demonstrated that isolated SNARE TMDs drastically lower the free energy of the stalk barrier and the metastable stalk (*Smirnova et al., 2019*). Yet, we note that exocytosis competence or priming of vesicles does not necessarily encompass hemifusion between the opposing membranes (*Figure 8*), but may represent any precursor state that is characterized by lowered intermembranous repulsion. Intracellular application of negative curvature promoting phospholipids like OA at least in part restored the reduced secretion rate seen with the polyL mutant (*Figure 6A–C*). This result is consistent with the view that cone-shaped phospholipids like OA promote the transition to the intermembrane stalk, which is characterized by net negative curvature (*Chernomordik and Kozlov, 2008*; *Kawamoto et al., 2015*) or may lower the energy for overcoming the hydration repulsion between the fusing membranes (*Smirnova et al., 2019*). Furthermore, intracellular application of curvature promoting phospholipids significantly altered synchronous secretion (*Figure 4—figure supplement 1*). The detailed kinetic analysis of the capacitance changes revealed that the rapidly releasable pool (RRP) was oppositely affected by OA and LPC, but no changes in exocytosis timing were observed. This indicates that curvature promoting phospholipids, like the polyL mutation (*Dhara et al., 2016*), regulate pool formation and/or exocytosis competence but do not interfere with exocytosis triggering (*Figure 8*). Collectively, these results suggest synergistic actions of lipids and SNARE-TMDs, which are critical for lowering the energy barrier to establish exocytosis competence.

Our findings contradict earlier observations that suggested elevated exocytosis by intracellular LPC and inhibitory effects by OA in response to depolarization-evoked stimulation of chromaffin cells (*Zhang and Jackson, 2010*). By using either photolytic uncaging of intracellular $Ca^{2+}$ (*Figure 4—figure supplement 1*) or $Ca^{2+}$ infusion (*Figure 4*), we have observed opposite effects if the curvature-mediating lipid components were applied intracellularly and found, in contrast to Zhang and Jackson, only a slight increase in the sustained rate of secretion with LPC (p=0.04, flash experiments, *Figure 3—figure supplement 1*) or no effect with their extracellular application in $Ca^{2+}$-

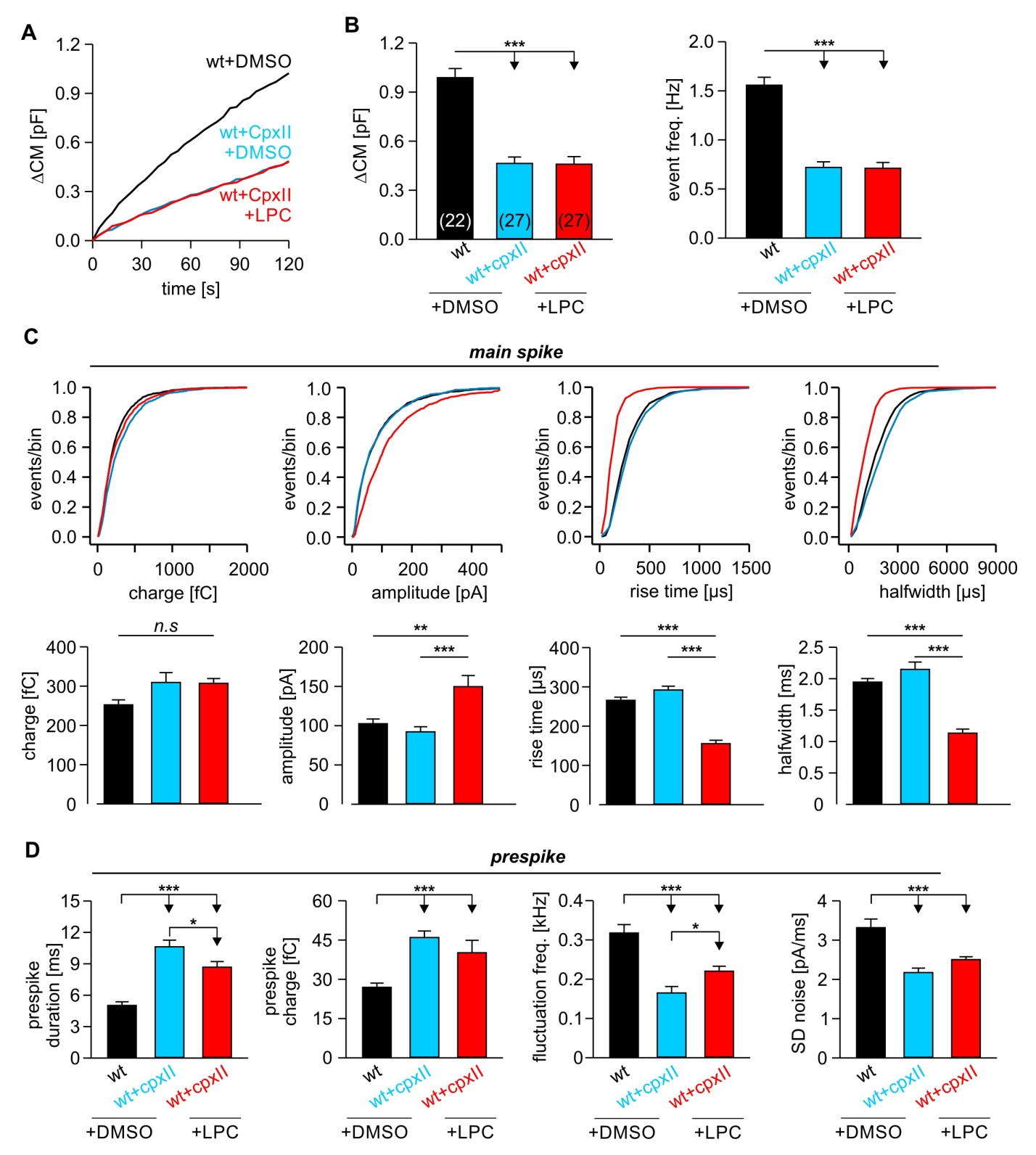

**Figure 7.** LPC fails to rescue the CpxII mediated prolongation in the expansion time of the nascent fusion pore. (**A**) Stimulus evoked averaged ΔCM for the indicated groups. (**B**) The CpxII mediated reduction of total ΔCM (left panel) and the amperometric event frequency (right panel) is unaffected by application of LPC. Numbers indicate analyzed cells. (**C**) Cumulative frequency distributions (top panels) and cell-weighted averages (bottom panels) show that extracellular LPC speeds up the kinetics of the main spike in CpxII expressing cells. (**D**) CpxII selectively slows down the kinetics of prespike

*Figure 7 continued on next page*

*Figure 7 continued*

phase. Application of LPC largely fails to restore the CpxII mediated deceleration of neurotransmitter release from single vesicle. Values are given as mean of median determined from the indicated parameter's frequency distribution for each cell. Data were collected from cells/events measured for wt +extra.DMSO (22/2171), wt+CpxII+extra.DMSO (27/995), and wt+CpxII+extra.LPC (27/1098). Only cells with >20 events were considered. Data are represented as mean ± SEM. *p<0.05, **p<0.01, ***p<0.001, one-way analysis of variance followed by Tukey-Kramer post test.

infusion experiments (*Figure 3*). Since LPC significantly influences the activation of voltage-dependent $Ca^{2+}$-channels (*Ben-Zeev et al., 2010*), potential changes in intracellular $[Ca^{2+}]i$ upon treatment of chromaffin cells with curvature-modifying lipid components may have masked their actual effect on membrane fusion, thus, providing an explanation for the apparently contradicting data. Alternatively, one might speculate that different exocytosis timing and intracellular $Ca^{2+}$ concentrations stimulating fusion in the depolarization-evoked response of chromaffin cells (*Zhang and Jackson, 2010*) and our $Ca^{2+}$-infusion experiments may contribute to these apparent discrepancies.

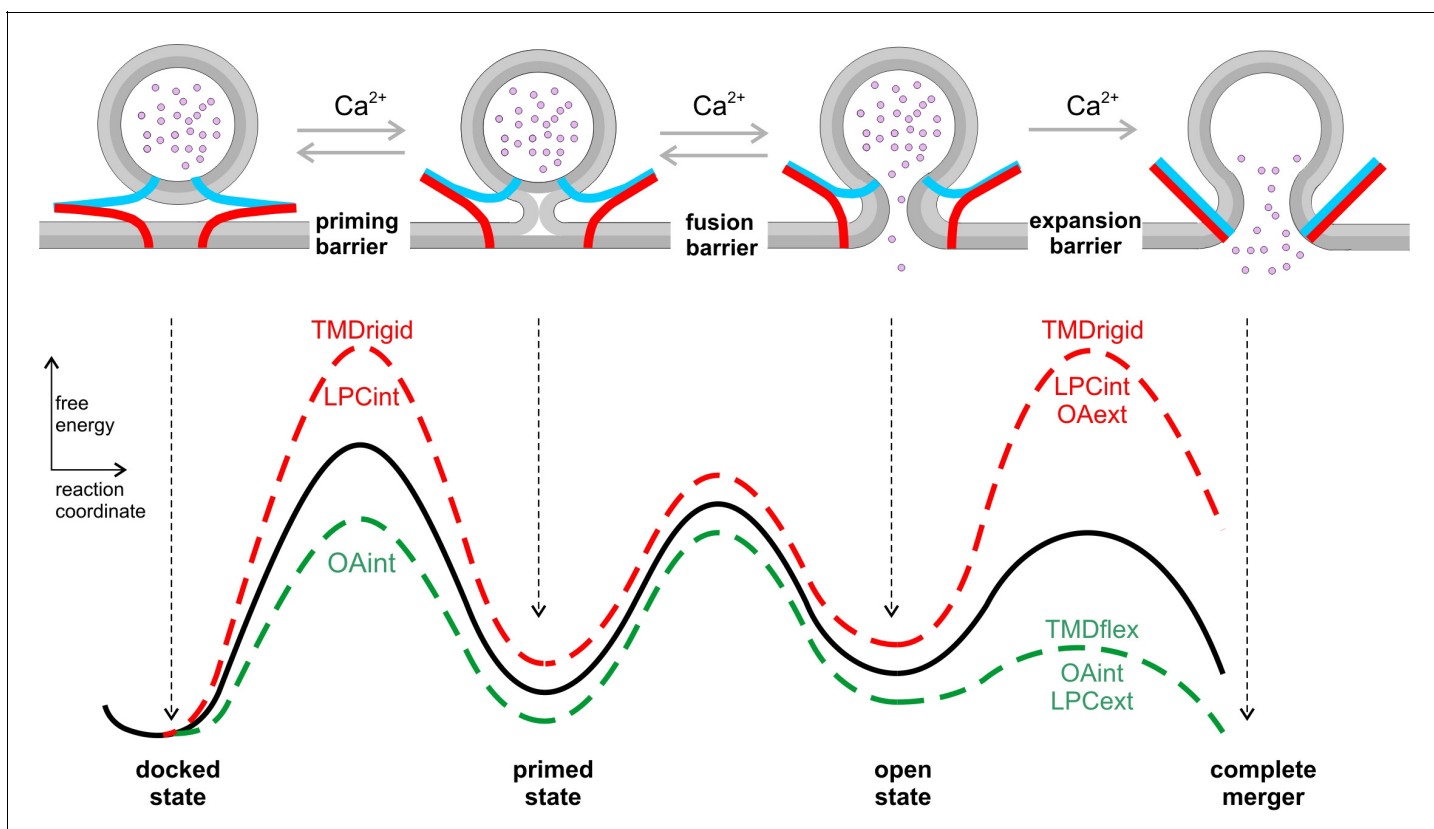

**Figure 8.** Hypothetical model of the energy landscape *en route* to membrane fusion. Vesicles states are depicted as corresponding to local minima in an energy profile. The vesicles transit from a docked to a primed state, to the nascent fusion pore state and finally into the state of complete membrane merger. Progressive SNARE zipping (VAMP2, blue; syntaxin, red; SNAP-25 not shown for clarity) together with $Ca^{2+}$ drives the forward reactions throughout the whole fusion process overcoming the energy barriers for priming and fusion up to the expansion of the fusion pore. TMD rigidity increases the energy for priming and also for pore expansion, whereas TMD flexibility profoundly lowers the latter. Intracellular OA rescues in part the secretion deficit of the VAMP2polyL mutant because the spontaneous curvature of the cytoplasmic leaflet is energetically significant in stalk formation. Neither TMD mutations nor curvature-modifying phospholipids affect the exocytosis timing. For pore expansion, instead, OA and LPC produce opposing effects in the same leaflet but similar actions when applied in different leaflets, consistent with membrane curvature-driven changes in the expansion kinetics of a purely lipidic pore. Notably, curvature accommodating phospholipids are expected to lower the energy of the semi-stable highly curved open pore state, but even more strongly reduce the energy barrier for pore growth (*Chizmadzhev et al., 2000*), thereby leveling of any differences in pore expansion between VAMP2 or its VAMP2polyL mutant.

## The role of v-SNARE TMDs in fusion pore dynamics

Above discussion does not provide an explanation for the question how v-SNARE TMDs alter subsequent fusion pore dynamics. A clue may come from recent biochemical studies showing that SNARE TMDs produce a negative hydrophobic mismatch by being too short with respect to their native membranes (*Milovanovic et al., 2015*). In the same line, molecular simulation detected local thinning of the membrane imposed by SNARE TMDs (*Smirnova et al., 2019*). Given that a negative lipid/peptide hydrophobic mismatch causes thinning/softening of membranes (*de Planque et al., 1998*; *Kim et al., 2012*; *Agrawal et al., 2016*) and even promotes the formation of inverted membrane phases (*de Planque and Killian, 2003*), it is possible that the concentration of SNARE TMDs at the stalk base or the pore rim leads to local changes in membrane curvature or elasticity, that favor fusion pore expansion. Such a scenario fits well with our previous observation that substitution of the VAMP2 TMD with a lipid anchor strongly decelerated kinetics of transmitter discharge, demonstrating the inherent propensity of the proteinaceous TMD to promote fusion pore expansion (*Dhara et al., 2016*). It also agrees with experiments in reduced model systems suggesting that lipidic SNARE-anchors largely fail in promoting proper fusion between artificial liposomes (*McNew et al., 2000*), cells expressing 'flipped' SNAREs (*Giraudo et al., 2005*), or between liposomes and lipid nanodiscs (*Shi et al., 2012*; *Bao et al., 2016*). Our new results now also show that a TMD-mediated deceleration in fusion pore dilation can be fully compensated by adding lipid molecules with a molecular shape that is favorable for the respective leaflet (*Figures 5* and *6*). This suggests that the actions of TMDs and curvature-generating lipids on the fusion process are similar, most likely reflecting their common ability to compensate for hydrophobic interstices associated with fusion intermediates. Notably, both tested lipid components, either OA or LPC, produce opposing effects on the cytoplasmic and extracellular leaflet, rendering the possibility unlikely that either lipid compound significantly flips between leaflets in the course of our experiments. Similar results with OA and LPC on fusion pore dynamics have previously been observed in secretion from chromaffin cells stimulated by membrane depolarizations (*Amatore et al., 2006*; *Zhang and Jackson, 2010*). Small fusion pores are highly curved lipidic structures that inherently exhibit regions of lipid packaging frustration as observed for membrane stalk intermediates (*Smirnova et al., 2019*). Consequently, curvature-accommodating phospholipids are able to facilitate pore enlargement in a leaflet specific manner (*Figures 3* and *4*). We hypothesize that the α-helical (rigid) polyL mutant would disfavor such highly bent regions whereas more flexible TMD variants should adapt better to such structures because structural flexibility allows them to explore a larger range of conformational space (*Figure 8*). The profound impact of LPC and OA on fusion pore expansion illustrates active remodeling of membranes which negates any thermodynamic differences between the actions of the TMD variants, explaining why the addition of LPC or OA overrides the influence of the v-SNARE TMDs. Notably, comparable interactions between differentially helical TMDs and hexadecane molecules were observed for viral fusion proteins (*Dennison et al., 2002*), revealing fundamental similarities in the mechanisms of $Ca^{2+}$-mediated and viral membrane fusion.

In contrast, extracellular LPC application was largely unable to reverse the delayed fusion pore expansion induced by CpxII expression (*Figure 7*). This observation conveys two important points: On one hand, it shows that curvature-accommodating phospholipids do not generally accelerate $Ca^{2+}$-triggered membrane fusion in a protein-independent fashion. On the other hand, it demonstrates that successful zipping of SNAREs, encompassing the step of lifting the CpxII clamp, is a prerequisite for accelerating fusion pore expansion by curvature modifying lipids. This observation highlights important constraints in the potential sequence of molecular events leading to fusion. It renders the possibility unlikely, that SNARE complex formation has been completed before fusion pore opening, while most of its energy is intermittently stored in bending stress of the stiff linker region (connecting SNARE motif and TMD) before being utilized in a mousetrap-like mechanism to open and expand the fusion pore. The close coupling between SNARE complex formation and membrane bending illustrates that assembling SNAREs actively drive and control the whole fusion process up to the expansion of the fusion pore (*Figure 8*).

Overall, our results provide new insight into the interplay of SNAREs and lipids in membrane fusion. They highlight how structural flexibility as a key feature of v-SNARE TMDs together with phospholipids guides SNARE-mediated exocytosis in a functional *pas de deux* towards fusion.

# Materials and methods

## Key resources table

| Reagent type (species) or resource | Designation | Source or reference | Identifiers | Additional information |
|---|---|---|---|---|
| Strain, strain background (*Mus musculus*) | C57BL/6 | | | |
| Genetic reagent (*Mus musculus*) | *VAMP2 – VAMP3* null allele | *Borisovska et al., 2005* | PMID:15920476 | |
| Antibody | mouse anti-VAMP2 | Synaptic Systems | Cat# 104 211 | ICC, 1:1000 |
| Antibody | rabbit anti-VAMP3 | Synaptic Systems | Cat# 104 103 | ICC, 1:1000 |
| Antibody | Alexa Fluor 555 goat anti-mouse | Invitrogen | Cat# A21422 | ICC: 1:1000 |
| Antibody | Alexa Fluor 488 goat anti-rabbit | Invitrogen | Cat# A11008 | ICC: 1:1000 |
| Recombinant DNA reagent (*Mus musculus*) | VAMP2 | GeneID: 22318 | cDNA (*Mus musculus*) | |
| Recombinant DNA reagent (*Mus musculus*) | VAMP1 | GeneID: 22317 | cDNA (*Mus musculus*) | |
| Recombinant DNA reagent (*Mus musculus*) | VAMP8 | GeneID: 22320 | cDNA (*Mus musculus*) | |
| Recombinant DNA reagent (*Mus musculus*) | CpxII | GenBank: U35101.1 | cDNA (*Mus musculus*) | |
| Transfected construct (*Mus musculus*) | pSFV-VAMP2-IRES-EGFP | this paper | | Semliki Forest virus derived from 22318, expression of wild type full length protein |
| Transfected construct (*Mus musculus*) | pSFV-VAMP1-IRES-EGFP | this paper | | Semliki Forest virus derived from 22317, expression of wild type full length protein |
| Transfected construct (*Mus musculus*) | pSFV-VAMP8-IRES-EGFP | this paper | | Semliki Forest virus derived from 22320, expression of wild type full length protein |
| Transfected construct (*Mus musculus*) | pSFV-VAMP2-poly L-IRES-EGFP | this paper | | Semliki Forest virus derived from 22318 with indicated mutations |
| Transfected construct (*Mus musculus*) | pSFV-VAMP2-VAMP1 TMD-IRES-EGFP | this paper | | Semliki Forest virus derived from 22318 and from 22317, chimeric mutant protein |
| Transfected construct (*Mus musculus*) | pSFV-VAMP2-VAMP 8TMD-IRES-EGFP | this paper | | Semliki Forest virus derived from 22318 and from 22320, chimeric mutant protein |

*Continued on next page*

*Continued*

| Reagent type (species) or resource | Designation | Source or reference | Identifiers | Additional information |
|---|---|---|---|---|
| Transfected construct (*Mus musculus*) | pSFV-CpxII-IRES-EGFP | this paper | | Semliki Forest virus derived from U35101.1; expression of wild type full length protein |
| Software algorithm | IgorPro | WaveMetrics Software | | |
| Software algorithm | AutesP | NPI electronics | | |
| Software algorithm | ImageJ | NIH | | |
| Software algorithm | Zen2008 | Zeiss | | |

## Culture of chromaffin cells

Experiments were performed on mouse chromaffin cells prepared at embryonic stage E17.5–E18.5 from double-v-SNARE knock-out mice (dko cells; *VAMP2*$^{-/-}$/*VAMP3*$^{-/-}$; *Borisovska et al., 2005*). Preparation of adrenal chromaffin cells was performed as described before (*Borisovska et al., 2005*). Recordings were done at room temperature on 1–3 days in culture (DIC) and 4.5–5.5 hr after infection of cells with virus particles.

Viral constructs cDNAs encoding for VAMP2 and its VAMP2-VAMP8- or VAMP2-VAMP1-TMD mutants were subcloned into the viral plasmid pSFV1 (Invitrogen, San Diego, CA), upstream of an internal ribosomal entry site (IRES) controlled open reading frame that encodes for enhanced green fluorescent protein (EGFP). EGFP expression (excitation wavelength 477 nm) served as a marker protein to identify infected cells. Mutant constructs were generated by PCR using the overlap expansion method (*Higuchi et al., 1988*). All mutations were verified by DNA sequence analysis (MWG Biotech, Germany). For the production of infectious virions, the cDNA was linearized with restriction enzyme SpeI and transcribed in vitro by using SP6 RNA polymerase (Ambion, USA). BHK21 cells were co-transfected by electroporation (400V, 975 µF) using a combination of 10 µg VAMP2 (wt or mutant) and pSFV-helper2 RNA. Virions released into the supernatant were harvested after 15 hr incubation (31°C, 5% $CO_2$) by low speed centrifugation (200 g, 5 min), snap-frozen and stored at −80°C (*Ashery et al., 1999*).

## Whole-cell capacitance measurements and amperometry of chromaffin cells

For extracellular application of curvature–inducing lipids, cells were incubated for 3 min in Ringer's solution containing 2 µM of either lysophosphatidylcholine (LPC) or oleic acid (OA) and 0.5% DMSO. Control cells were only treated with 0.5% DMSO. Immediately before the electrophysiological measurement cells were transferred to lipid and DMSO free Ringer's solution to facilitate gigaseal formation. For intracellular application, LPC or OA (dissolved in DMSO) was added to the intracellular solution at a final concentration of 5 µM and 0.5% DMSO. Whole-cell membrane capacitance measurements and photolysis of caged $Ca^{2+}$ as well as ratiometric measurements of $[Ca^{2+}]i$ were performed as described previously (*Borisovska et al., 2005*). The extracellular Ringer's solution contained (in mM): 130 NaCl, 4 KCl, 2 $CaCl_2$, 1 $MgCl_2$, 30 glucose, 10 HEPES-NaOH, pH 7.3, 320 mOsm. Ratiometric $[Ca]i$ measurements were performed using a combination of fura2 and furaptra (Invitrogen) excited at 340 nm and 380 nm. Whole-cell recordings with test solutions of defined calcium concentrations were used for in vivo-calibration of the ratiometric $Ca^{2+}$-signals. The intracellular solution for flash experiments contained (in mM): 110 Cs-glutamate, 8 NaCl, 3.5 $CaCl_2$, 5 NP-EGTA, 0.2 fura2, 0.3 furaptra, 2 MgATP, 0.3 $Na_2$GTP, 40 HEPES-CsOH, pH 7.3, 310 mOsm. The flash-evoked capacitance response was fitted with the function: $f(x)=A0 + A1(1-exp[-t/\tau 1]) + A2(1-exp[-t/\tau 2]) + kt$, where A0 represents the cell capacitance before the flash. The parameters A1, τ1, and A2, τ2, represent the amplitudes and time constants of the RRP and the SRP, respectively (*Rettig and Neher, 2002*). The secretory delay was defined as the time between the UV-flash and the intersection point of the back-extrapolated fast exponential with the baseline. Data were

acquired with the Pulse software (HEKA, Lambrecht, Germany) and capacitance measurements were performed according to the Lindau-Neher technique (sine wave stimulus: 1000 Hz, 35 mV peak-to-peak amplitude, DC-holding potential –70 mV). Current signals were digitized at 20 kHz and membrane capacitance was analyzed with customized IgorPro routines (Wavemetrics, Lake Oswego, OR).

Production of carbon fiber electrode (5 µm diameter, Amoco) and amperometric recordings with an EPC7 amplifier (HEKA Elektronik) were done as described before (*Bruns, 2004*). For $Ca^{2+}$ infusion experiments the pipette solution contained (in mM): 110 Cs-glutamate, 8 NaCl, 20 DPTA, 5 $CaCl_2$, 2 MgATP, 0.3 $Na_2GTP$, 40 HEPES-CsOH, pH 7.3, 310 mOsm (19 µM free calcium). Amperometric current signals were filtered at 2 kHz and digitized gap-free at 25 kHz. For the event frequency analysis, amperometric signals with a charge ranging from 10 to 5000 fC and peak amplitude >4 pA were selected. To determine single spike characteristics with reasonable fidelity, only events with an amplitude of >7 pA were analyzed. For the fluctuation and rms noise analyses prespike signals with durations longer than 2 ms were considered and the current derivative was additionally filtered at 1.2 kHz. Fluctuations exceeding the threshold of ±6 pA/ms (~4 times the average baseline noise) were counted. The fluctuation frequency was determined from the number of suprathreshold current fluctuations divided by the corresponding prespike signal duration.

### Immunocytochemistry

Chromaffin cells were processed 3.5 hr after virus infection for immunolabeling as described previously (*Borisovska et al., 2012*). For immunostaining the affinity-purified mouse monoclonal antibody against VAMP2 (clone 69.1, antigen epitope amino acid position 1–14, kindly provided by R. Jahn, MPI for Biophysical Chemistry, Göttingen, Germany) and/or a rabbit polyclonal antibody against VAMP3 (TG-21, Synaptic Systems, Göttingen Germany). Digital images (eight bit encoded) were acquired with an AxioCam MRm-CCD camera (Carl Zeiss, Inc) on the stage a Zeiss AxioVert 25 microscope using a 100x fluar oil-immersion objective (NA, 1.3, Zeiss, Göttingen, Germany). Images were analyzed with ImageJ software version 1.45. The total intensity of the fluorescent immune label was determined over the area of the outer cell circumference minus the area comprising the cell nucleus.

To study the subcellular localization and sorting of the mutant VAMP2 variants to large dense core vesicles, chromaffin cells were imaged with the ELYRA PS.1 superresolution microscope (Carl Zeiss). Images were acquired in the wide-field mode using a 63x Plan-Apochromat (NA 1.4) oil-immersion objective on the stage of a Zeiss Axio Observer at 488 and 561 nm wavelength of the excitation light and then processed for SIM with the Zen2012 software (Carl Zeiss). Z-stacks of 110 nm step size were used to identify the cell's footprint in order to minimize the contribution of ER/Golgi-derived fluorescence in virus-transfected cells. Cells were analyzed with the software package ImageJ, version 1.45. After threshold subtraction, the Mander's weighted colocalization coefficient was determined from the sum of VAMP2 pixels intensities that colocalizes with VAMP3, divided by the overall sum of VAMP2 pixels intensities (*Bolte and Cordelières, 2006*). Therefore, $M_{VAMP2} = \Sigma_{VAMP2}$ pixel intensity (coloc. VAMP3 pixel)/ $\Sigma_{VAMP2}$ pixel intensity (*Manders et al., 1993*). To determine the mean fluorescence intensity per granule, signals exceeding the threshold (~six standard deviation (SD) of average baseline noise) were analyzed.

### Statistical analysis

Values are given as mean ± SEM (standard error of mean) unless noted otherwise in the figure legends. To determine statistically significant differences, one-way analysis of variance and a Tukey–Kramer post hoc test were used, if not stated otherwise.

## Acknowledgements

The authors would like to express their gratitude J Rettig and D Stevens for valuable discussions. We thank W Frisch, P Schmidt, V Schmidt and M Roter for excellent technical assistance and E Krause for help with the SIM-microscopy (platform project of SFB894). The work was supported by grants from the DFG (SFB 1027) to DB and RM and by HOMFOR to MD.

## Additional information

### Funding

| Funder | Grant reference number | Author |
|---|---|---|
| Deutsche Forschungsge-meinschaft | SFB 1027 | Ralf Mohrmann Dieter Bruns |
| University of Saarland, Medical Faculty | HOMFOR | Madhurima Dhara |

The funders had no role in study design, data collection and interpretation, or the decision to submit the work for publication.

### Author contributions

Madhurima Dhara, Yvonne Schwarz, Conceptualization, Data curation, Formal analysis, Writing - original draft; Maria Mantero Martinez, Mazen Makke, Data curation, Formal analysis, Writing - original draft; Ralf Mohrmann, Conceptualization, Writing - review and editing; Dieter Bruns, Conceptualization, Resources, Supervision, Funding acquisition, Validation, Writing - original draft, Project administration

### Author ORCIDs

Madhurima Dhara (iD) https://orcid.org/0000-0001-7745-472X
Ralf Mohrmann (iD) http://orcid.org/0000-0001-9279-5071
Dieter Bruns (iD) https://orcid.org/0000-0002-2497-1878

### Decision letter and Author response

Decision letter https://doi.org/10.7554/eLife.55152.sa1
Author response https://doi.org/10.7554/eLife.55152.sa2

## Additional files

### Supplementary files

• Transparent reporting form

### Data availability

All data generated or analysed during this study are included in the manuscript and supporting files.

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
