## [Decision Letter]

**Acceptance summary:**

This work builds on a previous publication from the Bruns laboratory (Dhara et al., 2016), and provides new results regarding the importance of the flexibility of the transmembrane domains of synaptobrevins/VAMPs. This work shows that increased content of β-branched residues in the VAMP8 transmembrane domain and decreased content of β-branched residues in the synaptobrevin1 transmembrane domain have corresponding effects on amperometric spike and foot kinetics. However, these manipulations did not affect the size of the exocytotic burst in caged calcium experiments. The authors also investigated if lipids regulate fusion pore kinetics.

**Decision letter after peer review:**

Thank you for submitting your article "The functional pas de deux of v-SNARE transmembrane domains and lipids in membrane fusion" for consideration by *eLife*. Your article has been reviewed by three peer reviewers, and the evaluation has been overseen by a Reviewing Editor and Richard Aldrich as the Senior Editor. The reviewers have opted to remain anonymous.

The reviewers have discussed the reviews with one another and the Reviewing Editor has drafted this decision to help you prepare a revised submission as a Research Advance.

Summary:

This work builds on a previous publication from the Bruns laboratory (Dhara et al., 2016), and provides new results regarding the importance of the flexibility of the transmembrane domains of synaptobrevins/VAMPs. The authors study synaptobrevin2 constructs where the transmembrane domain is replaced by that of synaptobrevin1 or that of VAMP8. They show that the increased content of β-branched residues in the VAMP8 transmembrane domain and the decreased content of β-branched residues in the synaptobrevin1 transmembrane domain have the corresponding effects on amperometric spike and foot kinetics, which is an interesting finding and the potential physiological significance in terms of fusion pore dilation needs in different cell types provides a reasonable conclusion. However, these manipulations did not affect the size of the exocytotic burst in caged calcium experiments. The authors also investigate if lipids regulate fusion pore kinetics. The effects of LPC and OA on fusion pore kinetics largely confirm previous results but additional and contradictory results are presented on spike kinetics and exocytosis rates. There is a long-standing debate on the nature, lipidic or proteinaceous, of the fusion pore. The authors propose here both transmembrane domains and lipids act together.

However, considering that this work is closely related to the work by Dhara et al., 2016 and lacks the novelty we would expect for an independent Research Article, it will be treated as a Research Advance.

Essential revisions:

Much emphasis is set on "adaptation of TMD variants with distinct conformational flexibility" to ensure efficiency of release "from differentially sized vesicles". A more detailed analysis of this hypothesis would be interesting. For example, more insight into the content of β-branched amino acids in transmembrane domains versus vesicle size for additional vesicle SNAREs (i.e. all VAMPs – 1,2,3,4,5,7,8) would be useful.

Surprisingly, the size of the exocytotic burst in caged calcium experiments, which is not discussed. While in their past work, the authors reported a profound decrease of the RRP/SRP for the polyL construct, no change is observed for the Syb1 TM domain, which deserves a discussion.

The experiments with lysoPC and OA (Figure 5) repeat previous studies by Zhang and Jackson, 2010, which is cited by the authors. The authors state that their findings contradict those previous ones. There are some issues with the discussion of Zhang and Jackson, 2010. Zhang and Jackson studied the effects of LPC and OA in chromaffin cells, not in PC12 cells, as stated by the authors and the effects of intracellular LPC and OA were studied in patch clamp experiments with stimulation by 6s depolarization. It is not evident how these very different experiments are contradicted by the data presented here.

Zhang and Jackson, 2010 reported that extracellular LPC reduces foot duration whereas extracellular OA increased foot duration. Intracellular LPC and OA has the opposite effect. These results are fully consistent with the results reported in this manuscript and this should be clearly stated in the Discussion.

Zhang and Jackson reported an increase in spike rates with extracellular OA and a decrease in spike rates with extracellular LPC in chromaffin cells stimulated with high K. A small increase in exocytosis is also evident in Figure 3—figure supplement 1B (caged Ca^2+^). One question relates to the protocol used in the present manuscript. For intracellular dialysis with 19 µM free Ca^2+^ it is not clear how to interpret the data of panels such as Figure 4A and B. What is time zero in Figure 4A and similar panels and how much of the RRP/SRP is actually recorded here? Could this difference in protocols explain the apparent discrepancy with Zhang and Jackson, 2010 when it comes to exocytosis rates? Is there an effect of LPC/OA incubation time? Another difference is the adult rat vs. embryonal mouse dko cells with viral transfection of Syb2.

The results on the overall fusion kinetics triggered by calcium seem to be contradictory to their previous publication: the transmembrane mutations dramatically change the overall kinetics of membrane fusion in previous work (Figure 1 in Dhara et al., 2016), whereas no effect is observed in this work (Figure 2). Please explain.

The effect of curvature-generating lipids on membrane fusion has been extensively investigated. The results in this work seem to confirm the major conclusions from these previous results. Please discuss.

Similar effects are observed for alteration of v-SNARE transmembrane domains and lipids, likely suggesting a common underlying mechanism. Do the transmembrane domains generate curvature in a way similar to cone-shaped lipids? Or do the transmembrane domains affect the overall protein-membrane bending flexibility? The authors attribute the transmembrane domain effect to flexibility. It is unclear to what aspect of "flexibility" that the authors refer to. Moreover, are there explicit experimental data that show that transmembrane domains with more branched residues are more flexible?

It is unclear if cone-shaped lipids (like OA) fit the fusion stalk better than an inverted cone (like LPC), although this idea is widely found in the literature. The stalk has a negative Gaussian curvature, meaning opposing curvatures: while OA certainly fits better to the stalk along the pore direction from an intracellular side, it does not fit well to the stalk along the direction perpendicular to the neck. To see this more clearly, one may stretch the stalk without a change in pore diameter, the neck would approach a zero Gaussian curvature. In this case, PLC would be expected to stabilize the stalk. Given the uncertainty in the geometry of a stalk, it seems likely that OA and LPC can cooperatively stabilize the stalk. Thus, it is unclear why one is favored and the other is not.

Presentation:

In several places the authors assume that reader will have prior knowledge of the concepts and the authors' earlier work. This makes the text harder to follow. As an example, the content of β-branched aminoacids in TMDs is often related to "TMD flexibility" in general, but the directionality of the relationship is never directly stated. For example, higher number of β-branched amino acids confer more flexibility to the TMD of VAMP2. Some of this was explained in the earlier (Dhara et al., 2016), so please modify the Introduction accordingly by beginning to reference the earlier work explicitly at the beginning.

The main motivation for this study is to investigate the mechanism by which the SNARE transmembrane domains promote fusion. The authors point out correctly that recent work suggested an active role of SNARE transmembrane domains. However, they do not give credit to the publications that initially established the significance of the transmembrane domains such as McNew et al., 2000. They also need to put their work into perspective relating it other studies looking at the mechanism by which the SNARE TM domains promote fusion, such as Ngatchou et al., 2010 PNAS. Sharma and Lindau, 2018 PNAS, proposed a detailed mechanism for the function of the SNARE TMDs. How do the results presented here relate to that mechanism? Are they consistent with it or do they suggest otherwise?

The discussion on the role of v-SNARE transmembrane domains in fusion and in fusion pore dynamics should be made more concise and should specifically relate to vSNARE transmembrane domain roles that have been proposed in the literature. Does the study presented here support or invalidate any of the proposed mechanisms? The authors should also discuss how they explain the observed effects (or absence thereof) of LPC and OA on fusion rates.

[Editors' note: further revisions were suggested prior to acceptance, as described below.]

Thank you for resubmitting your work entitled "Synergistic actions of v-SNARE transmembrane domains and membrane-curvature modifying lipids in neurotransmitter release" for further consideration by *eLife*. Your revised Research Advance has been evaluated by Richard Aldrich (Senior Editor) and a Reviewing Editor.

The manuscript has been improved but there are two remaining issues that need to be addressed before acceptance, as outlined below:

1) The difference in protocols may contribute to the discrepancies between the current results and those of Zhang and Jackson, 2010. The effects of LPC and OA were investigated with very different protocols. Zhang and Jackson's experiments used 6s depolarization by KCl or whole cell patch clamp depolarization, with the highest fusion rates appearing early on while in the present manuscript whole cell infusion of buffered Ca^2+^ is applied with recording starting 3s after opening the cell. In addition to this difference in timing, the intracellular Ca^2+^ concentrations stimulating fusion events were presumably very different. These differences need to be discussed as possible cause of the apparent discrepancies.

2) Please add a discussion on the differential flexibility requirements of fusion pore dynamics and RRP/SRP support, which deserves a clear statement.

---

## [Author Response]

Essential revisions:Much emphasis is set on "adaptation of TMD variants with distinct conformational flexibility" to ensure efficiency of release "from differentially sized vesicles". A more detailed analysis of this hypothesis would be interesting. For example, more insight into the content of β-branched amino acids in transmembrane domains versus vesicle size for additional vesicle SNAREs (i.e. all VAMPs – 1,2,3,4,5,7,8) would be useful.

We agree with the reviewer’s point and this issue has been already described in detail in Table 1 of Dhara et al., 2016. Here, we showed that the content of β-branched amino acids within the N-terminal half of the VAMP2 (also known as synaptobrevin2) TMD systematically increases with the size of the secretory organelle from small over intermediate to large vesicles.

The previous results pointed to a hitherto unrecognized mechanism wherein TMDs of v-SNARE isoforms with a high content of β-branched amino acids are employed for efficient fusion pore expansion of larger sized vesicles, suggesting a general physiological significance of TMD flexibility in exocytosis.

To critically test this hypothesis, we have now comparatively analyzed the impact of chimeric TMD mutants of VAMP2 (exchanging its TMD with that of either VAMP8 or VAMP1) on the kinetics of transmitter release from chromaffin granules (Figure 1). In good agreement with our hypothesis, we now show that the VAMP8 TMD accelerates but the VAMP1 TMD decelerates transmitter release from single granules. Thus, not only artificial TMD mutants (carrying either helix-stabilizing leucines or flexibility–promoting β-branched isoleucine/valine residues) as studied by Dhara et al., 2016, but also naturally–occurring variants of v-SNARE TMDs specifically alter fusion pore dynamics.

Surprisingly, the size of the exocytotic burst in caged calcium experiments, which is not discussed. While in their past work, the authors reported a profound decrease of the RRP/SRP for the polyL construct, no change is observed for the Syb1 TM domain, which deserves a discussion.

We are grateful for the reviewer’s suggestion and have now discussed this topic in the revised manuscript (subsection “v-SNARE TMD variants differentially regulate membrane fusion”).

“Neither the decrease nor the increase of β-branched amino acids within the N-terminal half of the TMDs of VAMP1 and VAMP8 led to proportional changes in the secretion response when compared with VAMP2. For the VAMP2-VAMP1 TMD, it only slightly reduced the rate of tonic secretion in response to Ca^2+^-infusion (Figure 1A, B) and did not affect the synchronized, flash-evoked response (Figure 2). Thus, only large and systematic changes in the number of helix-rigidifying amino acids, like in the VAMP2polyL mutant, are able to interfere with exocytosis initiation.”

The experiments with lysoPC and OA (Figure 5) repeat previous studies by Zhang and Jackson, 2010, which is cited by the authors. The authors state that their findings contradict those previous ones. There are some issues with the discussion of Zhang and Jackson, 2010. Zhang and Jackson studied the effects of LPC and OA in chromaffin cells, not in PC12 cells, as stated by the authors and the effects of intracellular LPC and OA were studied in patch clamp experiments with stimulation by 6s depolarization. It is not evident how these very different experiments are contradicted by the data presented here.

We apologize for the mistake. Indeed, Zhang and Jackson used both, chromaffin cells and PC12 cells in their previous study (Zhang and Jackson, 2010). For intracellular application of phospholipids via the patch pipette depolarizations were employed, whereas for extracellular application KCl-mediated stimulation was used. We have now corrected this in the revised manuscript.

Yet, our results clearly counter those found by Zhang and Jackson (as stated already in the original manuscript):

“Our findings contradict earlier observations that suggested elevated exocytosis by intracellular LPC and inhibitory effects by OA in response to depolarization-evoked stimulation of chromaffin cells (Zhang and Jackson, 2010). By using either photolytic uncaging of intracellular Ca^2+^ (Figure 4—figure supplement 1) or Ca^2+^ infusion (Figure 4), we have observed opposite effects if the curvature-mediating lipid components were applied intracellularly and found, in contrast to Zhang and Jackson, only a slight increase in the sustained rate of secretion with LPC (p=0.04, flash experiments, Figure 3—figure supplement 1) or no effect with their extracellular application in Ca^2+^-infusion experiments (Figure 3).”

To provide a solution for these apparent discrepancies, we continued:

“Since LPC significantly influences the activation of voltage-dependent Ca^2+^-channels (Ben-Zeev et al., 2010), potential changes in intracellular [Ca^2+^]_i_ upon treatment of chromaffin cells with curvature-modifying lipid components may have masked their actual effect on membrane fusion, thus, providing an explanation for the apparently contradicting data.”

Given the suppressing effect of LPC on voltage-gated Ca^2+^-channels (Ben-Zeev et al., 2010), we on purpose stimulated exocytosis in chromaffin cells by either direct Ca^2+^-infusion or photolytic uncaging of intracellular Ca^2+^.

Zhang and Jackson, 2010 reported that extracellular LPC reduces foot duration whereas extracellular OA increased foot duration. Intracellular LPC and OA has the opposite effect. These results are fully consistent with the results reported in this manuscript and this should be clearly stated in the Discussion.

We have now stated theses similarities in the Discussion by adding the following text:

“Similar results with OA and LPC on fusion pore dynamics have previously been observed in secretion from chromaffin cells stimulated by membrane depolarizations (Amatore et al., 2006; Zhang and Jackson, 2010).”

Zhang and Jackson reported an increase in spike rates with extracellular OA and a decrease in spike rates with extracellular LPC in chromaffin cells stimulated with high K. A small increase in exocytosis is also evident in Figure 3—figure supplement 1B (caged Ca^2+^).

As pointed out above we found only a small increase in the sustained rate of secretion with LPC in photolytic Ca^2+^-uncaging experiments (Figure 3—figure supplement 1) or no effect of the curvature-modifying lipids on the exocytotic rates in Ca^2+^-infusion experiments (Figure 3). This result is consistent with the view that the spontaneous curvature of the cytoplasmic leaflet is energetically significant in intermembranous stalk formation, as stated in the legend of Figure 8.

One question relates to the protocol used in the present manuscript. For intracellular dialysis with 19 µM free Ca^2+^ it is not clear how to interpret the data of panels such as Figure 4A and B. What is time zero in Figure 4A and similar panels and how much of the RRP/SRP is actually recorded here? Could this difference in protocols explain the apparent discrepancy with Zhang and Jackson, 2010 when it comes to exocytosis rates?

We are grateful for the reviewer’s suggestion and have clarified this issue by adding the following information to the legend of Figure 1B: “T_0_ is the first time point of CM measurement, 2-3 sec after starting the Ca^2+^-infusion via the patch pipette.”

Given that chromaffin cells establish only a very small pool of primed vesicles under resting [Ca]_I_ levels, which depends on Ca^2+^-dependent priming reactions, not much of a RRP or SRP can be recorded just after opening the cell with the patch pipette (see also Rettig and Neher, 2002).

Is there an effect of LPC/OA incubation time?

We did not systematically study the effect of differential incubation time with LPC/OA on exocytosis. To minimize flipping of lipid molecules across the leaflets, we incubated cells for 3 min with the lipids and measured the cells within the next 20 min. As emphasized in the Discussion, OA and LPC, produce opposing effects on the cytoplasmic and extracellular leaflet, rendering the possibility unlikely that either lipid compound significantly flips between leaflets in the course of our experiments.

Another difference is the adult rat vs. embryonal mouse dko cells with viral transfection of Syb2.

We would not expect that the fundamental effects of curvature-modifying lipid molecules on membrane fusion differ among species like rat or mouse. Indeed, the effects of membrane curvature on fusion mechanics are largely similar from artificial lipid vesicles to secretory granules in cells (Chernomordik and Kozlov, 2003).

The results on the overall fusion kinetics triggered by calcium seem to be contradictory to their previous publication: the transmembrane mutations dramatically change the overall kinetics of membrane fusion in previous work (Figure 1 in Dhara et al., 2016), whereas no effect is observed in this work (Figure 2). Please explain.

This suggestion is probably due to a misunderstanding in the interpretation of our previous data. Our earlier experiments (Figure 1 in Dhara et al., 2016) have clearly shown that mutating the TMD core residues (amino acid positions 97-112) to helix-stabilizing leucines does not alter the rate constants of Ca^2+^-dependent release from RRP/SRP vesicles or the secretory delay, which are generally considered parameters describing stimulus-secretion coupling. In stark contrast, the size of SRP and RRP was dramatically reduced – but not to the exact same degree, resulting in the impression of a slight kinetic distortion. These results were interpreted as a fusion arrest of a population of vesicles caused by the PolyL mutant, but not as a triggering deficit. In our new dataset, we did not find any indication of a triggering effect, as rate constants were similarly unaffected for the chimeric TMD mutants. However, RRP and SRP were also unchanged, highlighting that the remaining flexibility of the chimeric TMD mutants is still sufficient to avert fusion arrest but does not allow pore evolution at normal velocity. Thus, there is no contradiction to the results reported previously.

The effect of curvature-generating lipids on membrane fusion has been extensively investigated. The results in this work seem to confirm the major conclusions from these previous results. Please discuss.

As mentioned above, we now explicitly state similarities, as well as discrepancies to previously published work in the Discussion of the revised manuscript.

Similar effects are observed for alteration of v-SNARE transmembrane domains and lipids, likely suggesting a common underlying mechanism. Do the transmembrane domains generate curvature in a way similar to cone-shaped lipids? Or do the transmembrane domains affect the overall protein-membrane bending flexibility?

We agree with the reviewer that these mechanistic questions are of central importance. As discussed in the original manuscript our results agree well with recent molecular dynamics simulations demonstrating that isolated SNARE TMDs drastically lower the free energy of the stalk barrier and the metastable stalk (Smirnova et al., 2019). Furthermore, our functional results are in excellent agreement with recent biochemical analyses, showing that conformationally rigid TMDs promote less proximal leaflet mixing and lipid splay than flexible TMDs (Scheidt et al., 2018). Therefore, an attractive explanation for this phenotype of the VAMP2polyL mutant would be that loss of conformational flexibility within the TMD lowers the probability of lipid splay and thereby impairs fusion initiation. We further note that intracellular application of negative curvature promoting phospholipids like OA at least in part restored the reduced secretion rate seen with polyL mutant (Figure 6A-C). This result is consistent with the view that cone-shaped phospholipids like OA promote the transition to the intermembrane stalk, which is characterized by net negative curvature (Chernomordik and Kozlov, 2008; Kawamoto et al., 2015) or may lower the energy for overcoming the hydration repulsion between the fusing membranes (Smirnova et al., 2019). Overall, these results suggest synergistic actions of lipids and SNARE-TMDs, which are critical for lowering the energy barrier to establish exocytosis competence.

With respect to the question how v-SNARE TMDs alter subsequent fusion pore dynamics, we pointed out in the Discussion that SNARE TMDs produce a negative hydrophobic mismatch by being too short with respect to their native membranes (Milovanovic et al., 2015). In the same line, molecular simulation detected local thinning of the membrane imposed by SNARE TMDs (Smirnova et al., 2019). Given that a negative lipid/peptide hydrophobic mismatch causes thinning/softening of membranes (de Planque et al., 1998; Kim et al., 2012; Agrawal et al., 2016) and even promotes the formation of inverted membrane phases (de Planque and Killian, 2003), it is possible that the concentration of SNARE TMDs at the stalk base or the pore rim leads to local changes in membrane curvature or elasticity, that favor fusion pore expansion. Such a scenario fits well with our previous observation that substitution of the VAMP2 TMD with a lipid anchor strongly decelerated kinetics of transmitter discharge, demonstrating the inherent propensity of the proteinaceous TMD to promote fusion pore expansion (Dhara et al., 2016). Strikingly, our new results now show that a TMD-mediated deceleration in fusion pore dilation can be largely eliminated by adding lipid molecules with a molecular shape that is favorable for the respective leaflet (Figures 5 and 6). This suggests that the actions of TMDs and curvature-generating lipids on the fusion process are similar, most likely reflecting their common ability to compensate for hydrophobic interstices associated with fusion intermediates. Further mechanistic studies are required to provide a more conclusive understanding of the fusion pore architecture. Taken together, we are convinced that we have thoroughly addressed the above questions in the discussion of the manuscript.

The authors attribute the transmembrane domain effect to flexibility. It is unclear to what aspect of "flexibility" that the authors refer to. Moreover, are there explicit experimental data that show that transmembrane domains with more branched residues are more flexible.

We thank the reviewer for their suggestion and elaborate on this topic more in the Introduction as follows:

“Our functional analysis supported the view that overall structural flexibility of the TMD, as promoted by the number of β-branched amino acids (like valine or isoleucine), rather than specific residues within the VAMP2 TMD, determines the exocytotic response (Dhara et al., 2016). Furthermore, molecular dynamics simulations of v-SNARE TMDs embedded in an asymmetric membrane (to mimic the physiological lipid composition of synaptic vesicles) revealed that mutant variants with a poly-leucine or with a poly-valine TMD decrease or increase the root mean square fluctuation (RMSF) of the backbone atoms for the peptide, respectively (Dhara et al., 2016). In the same line, sequence-specific back bone dynamics of isolated TMD model helices (probed by hydrogen/deuterium exchange) enhanced the fusogenicity of liposomes in in vitro assays (Stelzer et al., 2008; Quint et al., 2010), pointing to an active role of the v-SNARE TMD in the fusion mechanism.”

It is unclear if cone-shaped lipids (like OA) fit the fusion stalk better than an inverted cone (like LPC), although this idea is widely found in the literature. The stalk has a negative Gaussian curvature, meaning opposing curvatures: while OA certainly fits better to the stalk along the pore direction from an intracellular side, it does not fit well to the stalk along the direction perpendicular to the neck. To see this more clearly, one may stretch the stalk without a change in pore diameter, the neck would approach a zero Gaussian curvature. In this case, PLC would be expected to stabilize the stalk. Given the uncertainty in the geometry of a stalk, it seems likely that OA and LPC can cooperatively stabilize the stalk. Thus, it is unclear why one is favored and the other is not.

The energy to achieve hemifusion is sensitive to the monolayer’s curvature. As appreciated by the reviewer adding LPC to the contacting monolayers is a standard experimental procedure to inhibit hemifusion and has been shown to block fusion in a large variety of different systems (for review see Chernomordik and Kozlov, 2003). We also agree with the reviewer that for a roughly hourglass-shaped stalk, one surface curvature is positive, whereas the other is negative. The bending energy is expected to be at its minimum when the mean of the positive and the negative curvatures approaches the spontaneous curvature of the cytoplasmic monolayer. Accordingly the stalk will adjust its shape to minimize its energy (Markin and Albanesi, Biophys. J. 82:693-712, 2002), when the spontaneous curvature is changed by LPC. Yet, even with adjustment of the stalk shape, explicit calculation showed that the positive curvature of LPC will increase the stalk energy, consistent with fusion inhibition seen with inverted cone-shaped lipids (see Figure 8 in Kozlovsky et al., 2002, Biophys. J. 83, 2634-2651; Cohen and Melikan, J. Membr. Biol. 199, 1-14, 2004;).

Presentation:In several places the authors assume that reader will have prior knowledge of the concepts and the authors' earlier work. This makes the text harder to follow. As an example, the content of β-branched aminoacids in TMDs is often related to "TMD flexibility" in general, but the directionality of the relationship is never directly stated. For example, higher number of β-branched amino acids confer more flexibility to the TMD of VAMP2. Some of this was explained in the earlier (Dhara et al., 2016), so please modify the Introduction accordingly by beginning to reference the earlier work explicitly at the beginning.

We apologize for overlooking this point. As described above, we now state that TMD flexibility refers to the root mean square fluctuation (RMSF) of the backbone atoms for the peptide and that a higher number of β-branched amino acids confers more flexibility to the VAMP2 TMD, as shown by MD simulations (Dhara et al., 2016) as well as experimental data (Stelzer et al., 2008; Quint et al., 2010).

The main motivation for this study is to investigate the mechanism by which the SNARE transmembrane domains promote fusion. The authors point out correctly that recent work suggested an active role of SNARE transmembrane domains. However, they do not give credit to the publications that initially established the significance of the transmembrane domains such as McNew et al., 2000. They also need to put their work into perspective relating it other studies looking at the mechanism by which the SNARE TM domains promote fusion, such as Ngatchou et al., 2010 PNAS. Sharma and Lindau, 2018 PNAS, proposed a detailed mechanism for the function of the SNARE TMDs. How do the results presented here relate to that mechanism. Are they consistent with it or do they suggest otherwise?

As requested, we now cite previous work on the significance of TMDs in membrane fusion by adding the following text:

“It also agrees with experiments in reduced model systems suggesting that lipidic SNARE-anchors largely fail in promoting proper fusion between artificial liposomes (McNew et al., 2000), cells expressing ‘flipped’ SNAREs (Giraudo et al., 2005), or between liposomes and lipid nanodiscs (Bao et al., 2015; Shi et al., 2012).”

The polar C-terminal residues of the VAMP2 and syntaxin 1A transmembrane domains have been suggested to form a hydrophilic core between the two distal leaflets, thereby inducing fusion pore formation (Sharma and Lindau, 2018, PNAS, 115, 12751–12756). In view of these results, we intentionally did not mutate these C-terminal residues in order not to disturb the opening of the fusion pores. Our experiments therefore neither contradict nor confirm previous findings by Lindau and colleagues.

The discussion on the role of v-SNARE transmembrane domains in fusion and in fusion pore dynamics should be made more concise and should specifically relate to vSNARE transmembrane domain roles that have been proposed in the literature. Does the study presented here support or invalidate any of the proposed mechanisms? The authors should also discuss how they explain the observed effects (or absence thereof) of LPC and OA on fusion rates.

With the additions specified above, we feel that our revised discussion explicitly addresses how v-SNARE TMD variants differentially regulate transmitter discharge from single vesicles as well as how v-SNARE TMDs promote membrane fusion and fusion pore dynamics.

As requested, we now clearly state:

“Yet, the observed dependency of fusion rates on intracellular LPC and OA is difficult to reconcile with concepts that exocytosis begins with a proteinaceous fusion pore (Zhang and Jackson, 2010). They rather suggest that curvature-accommodating phospholipids facilitate stalk formation, catalyze the transition to pore opening and promote pore enlargement, results which are clearly in accord with the continuum ‘stalk-pore’ model of membrane fusion (Chizmadzhev et al., 1995; Chizmadzhev et al., 2000; Chernomordik and Kozlov, 2003).”

[Editors' note: further revisions were suggested prior to acceptance, as described below.]

The manuscript has been improved but there are two remaining issues that need to be addressed before acceptance, as outlined below:1) The difference in protocols may contribute to the discrepancies between the current results and those of Zhang and Jackson, 2010. The effects of LPC and OA were investigated with very different protocols. Zhang and Jackson's experiments used 6s depolarization by KCl or whole cell patch clamp depolarization, with the highest fusion rates appearing early on while in the present manuscript whole cell infusion of buffered Ca^2+^ is applied with recording starting 3s after opening the cell. In addition to this difference in timing, the intracellular Ca^2+^ concentrations stimulating fusion events were presumably very different. These differences need to be discussed as possible cause of the apparent discrepancies.

To accommodate the reviewer’s suggestion, we have added the following text:

“Alternatively, one might speculate that different exocytosis timing and intracellular Ca^2+^ concentrations stimulating fusion in the depolarization-evoked response of chromaffin cells (Zhang and Jackson, 2010) and our Ca^2+^-infusion experiments may contribute to these apparent discrepancies.”

2) Please add a discussion on the differential flexibility requirements of fusion pore dynamics and RRP/SRP support, which deserves a clear statement.

We now state:

“In comparison to the VAMP2polyL mutant, naturally occurring TMD variants change exocytosis competence (i.e. buildup of RRP/SRP) less effectively than they alter fusion pore dynamics. This behavior ensures bona fide transmitter release from differentially-sized vesicles without compromising their exocytosis competence.”